# GIST: Targeted Data Selection for Instruction Tuning via Coupled Optimization Geometry

**Guanghui Min** [1]  **Tianhao Huang** [1]  **Ke Wan** [1]  **Chen Chen** [1]

## Abstract

Targeted data selection has emerged as a crucial paradigm for efficient instruction tuning, aiming to identify a small yet influential subset of training examples for a specific target task. In practice, influence is often measured through the effect of an example on parameter updates. To make selection scalable, many approaches leverage optimizer statistics (e.g., Adam states) as an axis-aligned surrogate for update geometry (i.e., diagonal precondition), implicitly treating parameters as coordinate-wise independent. We show that this assumption breaks down in parameter-efficient fine-tuning (PEFT) methods such as LoRA. In this setting, the induced optimization geometry exhibits strong cross-parameter coupling with non-trivial off-diagonal interactions, while the task-relevant update directions are confined to a low-dimensional subspace. Motivated by this mismatch, we propose `GIST` (**G**radient **I**sometric **S**ubspace **T**ransformation), a simple yet principled alternative that replaces axis-aligned scaling with robust subspace alignment. `GIST` recovers a task-specific subspace from validation gradients via singular value decomposition (SVD), projects training gradients into this coupled subspace, and scores examples by their alignment with target directions. Extensive experiments have demonstrated that `GIST` matches or outperforms the state-of-the-art baseline with only 0.29% of the storage and 25% of the computational time under the same selection budget. Our code is available at https://github.com/GuanghuiMin/GIST.

[1]Department of Computer Science, University of Virginia, Charlottesville, USA. Correspondence to: Guanghui Min <jjm8vr@virginia.edu>.

*Proceedings of the 43rd International Conference on Machine Learning*, Seoul, South Korea. PMLR 306, 2026. Copyright 2026 by the author(s).

## 1. Introduction

Instruction tuning has become the de facto standard for aligning Large Language Models (LLMs) with human intent (Touvron et al., 2023; Ouyang et al., 2022). While early approaches relied on scaling up the volume of instruction data, recent findings have precipitated a paradigm shift: the quality and relevance of data are far more critical than mere quantity. Seminal works like LIMA (Zhou et al., 2023) and AlpaGasus (Chen et al., 2024) demonstrate that a small, carefully curated subset of high-quality examples can rival or even surpass the performance of models trained on massive, unfiltered datasets. This "less is more" phenomenon has catalyzed a surge of interest in automated data selection, aiming to identify the most effective training samples from large-scale pools to maximize downstream performance (Liu et al., 2024; Zhang et al., 2025).

Beyond general alignment, a more practical and challenging problem is **Targeted Instruction Tuning** (Xia et al., 2024), where the goal is to select a data subset that maximizes performance on a specific target distribution under a limited budget. Existing approaches generally fall into three categories. First, hard example mining prioritizes training samples based on scalar metrics like perplexity or loss magnitude (Zhao et al., 2024a; Yin & Rush, 2025; Antonello et al., 2021). Second, similarity-based methods utilize embedding retrieval to identify semantically relevant examples in the representation space (Zhang et al., 2018; Hanawa et al., 2020; Ivison et al., 2025). Finally, optimizer-based methods, exemplified by the state-of-the-art framework LESS (Xia et al., 2024), quantify data contribution directly in the gradient space via influence functions (Koh & Liang, 2017). Notably, LESS incorporates optimization geometry by leveraging Adam states, offering a scalable solution for LLMs.

However, relying on first-order optimizer statistics imposes a fundamental geometric limitation, prioritizing computational efficiency over the precision of the quadratic approximation. Designed for linear scalability, optimizers like Adam approximate the local curvature, referring to the Hessian geometry describing the loss landscape, via a *diagonal* preconditioner (Kingma & Ba, 2015). While efficient, this imposes a hard expressivity bottleneck: *diagonal matrices*

*lack the off-diagonal terms necessary to represent **parameter interactions** (e.g., the bilinear coupling in LoRA)*. Consequently, this structural incapacity inevitably distorts the intrinsic metric (Amari, 1998) of the parameter space. Furthermore, Adam's element-wise inversion indiscriminately amplifies directions with small gradients. In the context of low-rank adaptation, where the vast majority of the parameter space constitutes a noisy null space (Aghajanyan et al., 2021), this scaling boosts noise rather than preserving the sparse update structure. Ultimately, unlike Newton's method which leverages full curvature to identify optimal descent paths, this approximation fails to capture the true *descent potential* of induced updates, fundamentally misestimating their impact on loss reduction.

Motivated by this mismatch, we introduce GIST (**G**radient **I**sometric **S**ubspace **T**ransformation), a data selection framework that prioritizes geometric alignment over unstable diagonal approximations. GIST extracts a low-rank, task-specific subspace from validation gradients and scores training examples by projected gradient alignment, yielding an efficient selection rule that naturally respects parameter coupling without requiring full second-order information. Our contributions are threefold:

- **Theoretical Unification & Analysis:** We unify prior targeted selection methods as approximations to a common geometry-aware objective, revealing that diagonal preconditioners are inherently limited under rotated low-rank coupling. We then derive a principled, tractable non-diagonal estimator using the spectral structure of target gradients.
- **GIST Algorithm:** We introduce a scalable subspace-based selection method: GIST extracts a low-rank task subspace from validation gradients via SVD and ranks examples using projected gradient alignment, enabling efficient selection under coupled PEFT geometry.
- **Empirical Superiority:** Extensive experiments on MMLU (Hendrycks et al., 2020), TYDIQA (Clark et al., 2020), and BBH (Suzgun et al., 2023) demonstrate that GIST matches or outperforms the state-of-the-art baseline with only 0.29% of the storage and 25% of the computational time under the same selection budget.

## 2. Preliminaries

**Notations.** We denote the large-scale instruction tuning dataset as $\mathcal{D} = \{z_i\}_{i=1}^{|\mathcal{D}|}$ with $z_i = (x_i, y_i)$. The Large Language Model (LLM) is denoted as $\mathcal{M}_{\theta}$. We distinguish between the *sample-wise loss* on a data instance $z = (x, y)$, denoted as $\ell(z, \theta)$, and the *empirical risk* over a dataset $S$, denoted as $\mathcal{L}(S, \theta) = \frac{1}{|S|} \sum_{z \in S} \ell(z, \theta)$. All notations and symbols are summarized in Table 5.

### 2.1. Problem Definition

We formalize the targeted data selection task as a constrained bi-level optimization problem. Our goal is to select a subset $\mathcal{S} \subset \mathcal{D}$ with a cardinality constraint $|\mathcal{S}| = k$ (where $k \ll |\mathcal{D}|$ is the budget), such that the LLM trained on $S$ minimizes the loss on the target distribution $\mathcal{D}_{\text{val}}$, following the setting in Xia et al. (2024). Formally, this is defined as:

**Problem 1** (Targeted Data Selection). *Find the optimal subset $\mathcal{S}^*$ that solves the following bi-level optimization:*

$$\min_{S \subseteq \mathcal{D}} \quad \mathcal{L}(\mathcal{D}_{val}, \theta_S^*)$$
$$s.t. \quad |S| = k, \tag{1}$$
$$\theta_S^* = \arg\min_{\theta} \mathcal{L}(S, \theta).$$

### 2.2. Optimization Dynamics of Instruction Tuning

We analyze the optimization trajectory by examining the parameter update from step $t$ to $t+1$. For clarity, we consider a per-sample (batch size $= 1$) update, and denote the instantaneous training objective at step $t$ by $\ell(z_t, \theta)$. We study the local behavior of $\ell(z_t, \theta)$ around the current parameters $\theta_t$.

**Newton's Method.** Ideally, the parameter update is derived by minimizing a local quadratic approximation of the instantaneous loss. We perform a second-order Taylor expansion of $\ell(z_t, \theta)$ around $\theta_t$:

$$\ell(z_t, \theta) \approx \ell(z_t, \theta_t) + \mathbf{g}_t^\top \Delta\theta + \frac{1}{2}\Delta\theta^\top \mathbf{H}_t \Delta\theta, \quad (2)$$

where $\mathbf{g}_t \triangleq \nabla_{\theta}\ell(z_t, \theta_t)$ and $\mathbf{H}_t \triangleq \nabla_{\theta}^2\ell(z_t, \theta_t)$ denote the gradient and Hessian, respectively. Minimizing this quadratic surrogate yields the stationarity condition $\mathbf{g}_t + \mathbf{H}_t\Delta\theta = 0$ (assuming $\mathbf{H}_t$ is locally invertible or appropriately regularized). Solving for $\theta$ yields the classical Newton update step:

$$\theta_{t+1} = \theta_t - \eta\mathbf{H}_t^{-1}\mathbf{g}_t, \tag{3}$$

where $\eta$ is the learning rate. Here, $-\mathbf{H}_t^{-1}\mathbf{g}_t$ is the minimizer of the local quadratic surrogate, i.e., the **optimal second-order direction** under this approximation.

**Adam Optimizer.** In practice, LLMs are fine-tuned via the Adam optimizer (Kingma & Ba, 2015), utilizing exponential moving averages of gradient moments for adaptive updates. Structurally, the inverse second-moment term functions as a *diagonal surrogate* for the curvature matrix to precondition the optimization steps; we provide the detailed update formulations in Appendix D.

## 3. Theoretical Analysis

In this section, we analyze the data selection problem from the perspective of influence functions and optimization ge-

ometry. We first establish a unified view connecting data selection to gradient preconditioning. We then scrutinize standard approximations in prior work, specifically diagonal preconditioners from optimizer states, and demonstrate their theoretical failure modes in data selection. Finally, we derive a principled, tractable non-diagonal estimator based on the spectral structure revealed by target gradients.

### 3.1. A Unified Optimization View of Data Selection

We aim to select a training sample $z \in \mathcal{D}$ such that training on it maximizes the reduction in the validation risk $\mathcal{L}(\mathcal{D}_{\mathrm{val}}, \boldsymbol{\theta})$. To quantify the effect of one-step parameter update on $\mathcal{D}_{\mathrm{val}}$, we perform a first-order Taylor expansion of the validation loss $\mathcal{L}(\mathcal{D}_{\mathrm{val}}, \boldsymbol{\theta})$ around the current parameters $\boldsymbol{\theta}_t$, with $\Delta\boldsymbol{\theta}_t = \boldsymbol{\theta}_{t+1} - \boldsymbol{\theta}_t$:

$$\mathcal{L}(\mathcal{D}_{\mathrm{val}}, \boldsymbol{\theta}_{t+1}) \approx \mathcal{L}(\mathcal{D}_{\mathrm{val}}, \boldsymbol{\theta}_t) + \nabla_{\boldsymbol{\theta}}\mathcal{L}(\mathcal{D}_{\mathrm{val}}, \boldsymbol{\theta}_t)^{\top}\Delta\boldsymbol{\theta}_t. \quad (4)$$

Ideally, using a Hessian-preconditioned local update as in Koh & Liang (2017), the one-step change of the validation loss can be approximated by substituting a preconditioned update $\Delta\boldsymbol{\theta}_t$ into the first-order expansion. In particular, taking $\Delta\boldsymbol{\theta}_t \approx -\eta\, \mathbf{H}_{\mathrm{val},t}^{\dagger}\nabla_{\boldsymbol{\theta}}\ell(\boldsymbol{z}, \boldsymbol{\theta}_t)$ yields

$$\begin{aligned} \Delta\mathcal{L}_{\mathrm{val}}(\boldsymbol{z}) &= \mathcal{L}(\mathcal{D}_{\mathrm{val}}, \boldsymbol{\theta}_{t+1}) - \mathcal{L}(\mathcal{D}_{\mathrm{val}}, \boldsymbol{\theta}_t) \\ &\approx -\eta\nabla_{\boldsymbol{\theta}}\mathcal{L}(\mathcal{D}_{\mathrm{val}}, \boldsymbol{\theta}_t)^{\top}\mathbf{H}_{\mathrm{val},t}^{\dagger}\nabla_{\boldsymbol{\theta}}\ell(\boldsymbol{z}, \boldsymbol{\theta}_t), \quad (5) \end{aligned}$$

where $\mathbf{H}_{\mathrm{val},t} \triangleq \nabla_{\boldsymbol{\theta}}^2\mathcal{L}(\mathcal{D}_{\mathrm{val}}, \boldsymbol{\theta}_t)$. [1] This matrix characterizes the instantaneous curvature of the target task manifold. For a subset $S \subseteq \mathcal{D}$, we add the per-sample contributions due to linearity of empirical risk, yielding:

$$\begin{aligned} \Delta\mathcal{L}_{\mathrm{val}}(S) &\approx \sum_{\boldsymbol{z} \in S} \Delta\mathcal{L}_{\mathrm{val}}(\boldsymbol{z}) \\ &\propto -\nabla_{\boldsymbol{\theta}}\mathcal{L}(\mathcal{D}_{\mathrm{val}}, \boldsymbol{\theta}_t)^{\top}\mathbf{H}_{\mathrm{val},t}^{\dagger}\nabla_{\boldsymbol{\theta}}\mathcal{L}(S, \boldsymbol{\theta}_t). \quad (6) \end{aligned}$$

**Theorem 3.1** (Single-level Approximate Optimization). *Fix the current parameters $\boldsymbol{\theta}_t$. Under the first-order approximation, Problem 1 is approximately reduced to maximizing the predicted reduction in validation loss:*

$$\begin{aligned} \max_{S \subseteq \mathcal{D}} \quad & \nabla_{\boldsymbol{\theta}}\mathcal{L}(\mathcal{D}_{val}, \boldsymbol{\theta}_t)^{\top}\mathbf{H}_{val,t}^{\dagger}\nabla_{\boldsymbol{\theta}}\mathcal{L}(S, \boldsymbol{\theta}_t) \\ s.t. \quad & |S| = k, \end{aligned} \quad (7)$$

*up to constants independent of $S$ and higher-order terms.*

A full proof is provided in Appendix E.1. Theorem 3.1 reveals a geometric view of targeted data selection: each

---

[1] When $\mathbf{H}_{\mathrm{val},t}$ is singular or ill-conditioned (typical for over-parameterized models), we use the Moore–Penrose pseudoinverse $\mathbf{H}_{\mathrm{val},t}^{\dagger}$ (which is same with $\mathbf{H}_{\mathrm{val},t}^{-1}$ when $\mathbf{H}_{\mathrm{val},t}$ is invertible). Equivalently, $\mathbf{H}_{\mathrm{val},t}^{\dagger}$ yields the minimum-norm solution to $\mathbf{H}_{\mathrm{val},t}\Delta\boldsymbol{\theta} = -\nabla_{\boldsymbol{\theta}}\ell(\boldsymbol{z}, \boldsymbol{\theta}_t)$ and ignores directions in the null space.

candidate subset $S$ is scored by alignment under the metric induced by $\mathbf{H}_{\mathrm{val},t}^{\dagger}$ between the target gradient $\nabla_{\boldsymbol{\theta}}\mathcal{L}(\mathcal{D}_{\mathrm{val}}, \boldsymbol{\theta}_t)$ and the subset gradient $\nabla_{\boldsymbol{\theta}}\mathcal{L}(S, \boldsymbol{\theta}_t)$.

In modern LLM fine-tuning, however, forming $\mathbf{H}_{\mathrm{val},t}$ (let alone $\mathbf{H}_{\mathrm{val},t}^{\dagger}$) is intractable. Consequently, existing selection criteria can be viewed as optimizing Eq. (7) under different structural surrogates of $\mathbf{H}_{\mathrm{val},t}^{\dagger}$ and/or simplified assumptions about the gradient geometry, which we formalize next.

**Hard Example Mining (Assumption: Magnitude Prior).** Let $\tilde{\mathbf{g}}_{\mathrm{val},t} \triangleq (\mathbf{H}_{\mathrm{val},t}^{\dagger})^{1/2}\nabla_{\boldsymbol{\theta}}\mathcal{L}(\mathcal{D}_{\mathrm{val}}, \boldsymbol{\theta}_t)$ and $\tilde{\mathbf{g}}_t(S) \triangleq (\mathbf{H}_{\mathrm{val},t}^{\dagger})^{1/2}\nabla_{\boldsymbol{\theta}}\mathcal{L}(S, \boldsymbol{\theta}_t)$. Then we have

$$\begin{aligned} \nabla_{\boldsymbol{\theta}}\mathcal{L}(\mathcal{D}_{\mathrm{val}})^{\top}\mathbf{H}_{\mathrm{val},t}^{\dagger}\nabla_{\boldsymbol{\theta}}\mathcal{L}(S, \boldsymbol{\theta}_t) &= \tilde{\mathbf{g}}_{\mathrm{val}}^{\top}\tilde{\mathbf{g}}_S \\ &= \|\tilde{\mathbf{g}}_{\mathrm{val}}\|_2\, \|\tilde{\mathbf{g}}_S\|_2 \cos\angle(\tilde{\mathbf{g}}_{\mathrm{val}}, \tilde{\mathbf{g}}_S). \end{aligned} \quad (8)$$

Hard-example mining *implicitly assumes* that the directional term $\cos\angle(\tilde{\mathbf{g}}_{\mathrm{val}}, \tilde{\mathbf{g}}_S)$ varies little across candidate subsets (or is weakly informative), so maximizing the influence is approximated by maximizing the magnitude $\|\tilde{\mathbf{g}}_S\|_2$. In practice, hard-mining ranks individual samples by length of tokens (Zhao et al., 2024a) or perplexity (Yin & Rush, 2025) as a proxy for $\|\mathbf{g}(\boldsymbol{z})\|_2$, and then forms $S$ from top-ranked samples. For cross-entropy losses, low predicted probability on the ground-truth token(s) yields both larger loss and larger output-layer residuals, which typically increases the back-propagated gradient norm under mild smoothness/bounded-Jacobian conditions.

**Similarity-based Methods (Assumption: Representation Kernel Matching).** Embedding- or retrieval-based methods like RDS (Zhang et al., 2018; Hanawa et al., 2020) and RDS+ (Ivison et al., 2025) *implicitly assume* that the utility/influence in Eq. (7) can be well-approximated by a similarity kernel defined on a chosen representation (e.g., last-layer hidden states or external embedding models). Equivalently, they replace the $\mathbf{H}_{\mathrm{val},t}^{\dagger}$-induced parameter-space alignment in Eq. (7) with a representation-space kernel score and select nearest neighbors of $\mathcal{D}_{\mathrm{val}}$ under this kernel. This bypasses explicit modeling of the parameter-space metric induced by $\mathbf{H}_{\mathrm{val},t}^{\dagger}$ and induces a geometry determined by the chosen representation/kernel, which may miss task-relevant anisotropy present in parameter space.

**Optimizer-based Methods (Assumption: Locally Orthogonal Parameter Coordinates).** Scalable influence approximations, such as LESS (Xia et al., 2024), *implicitly assume* that the local metric induced by $\mathbf{H}_{\mathrm{val},t}^{\dagger}$ is approximately diagonal in the chosen parameter coordinates, i.e., cross-parameter couplings are negligible and the coordinates are (locally) orthogonal under this metric. Concretely, they use a diagonal surrogate of the form $\mathrm{diag}(\sqrt{\hat{\mathbf{v}}_t} + \epsilon)^{-1}$ in Eq. (22) derived from optimizer statistics (e.g., Adam's second-moment estimates). While this captures element-wise rescaling, it discards off-diagonal correlations in $\mathbf{H}_{\mathrm{val},t}^{\dagger}$

and thus cannot represent cross-parameter interactions encoded by curvature, which can change the ranking of utilities when off-diagonal correlations are non-negligible.

## 3.2. The Limitation: Diagonal Surrogates Cannot Handle Coupling

State-of-the-art methods like LESS (Xia et al., 2024) typically approximate the optimization geometry using diagonal preconditioners derived from optimizer states (e.g., Adam). This approach implicitly assumes that parameters are coordinate-wise independent.

Figure 1 shows that the validation-gradient matrix concentrates most of its variance in a low-dimensional principal subspace (rapid spectral decay and early saturation of explained variance), a consequence of the targeted setting where $\text{rank}(\mathbf{G}_{\text{val}}) \leq |\mathcal{D}_{\text{val}}| \ll d$. Importantly, *low-rank does not imply axis-alignment*. The resulting principal directions are generally *linear combinations* of coordinates, i.e., a rotated subspace that encodes genuine parameter coupling.

We show that coupling in PEFT is structural rather than incidental. Under LoRA (Hu et al., 2022) with $W = W_0 + BA$, even a local quadratic model in $W$ produces cross-block curvature between the LoRA factors $A$ and $B$ (Theorem 3.2).

**Theorem 3.2** (LoRA induces cross-block curvature). *Let $\mathcal{L}(W)$ be twice differentiable and consider the LoRA parameterization $W = W_0 + BA$, where $B \in \mathbb{R}^{m \times r}$ and $A \in \mathbb{R}^{r \times n}$. Let $\mathbf{G}_W = \nabla_W \mathcal{L}(W)$ and let $\mathbf{H}_W$ denote the Hessian with respect to $\text{vec}(W)$. For any $i \in [m]$, $j \in [n]$, and $k, k' \in [r]$,*

$$\frac{\partial^2 \mathcal{L}}{\partial B_{ik'} \partial A_{kj}} = \left\langle \mathbf{H}_W[B_{:k} e_j^\top], e_i A_{k':} \right\rangle_F + \delta_{kk'} (\mathbf{G}_W)_{ij}. \tag{9}$$

*Thus, the LoRA-parameter Hessian contains explicit cross-block terms between $A$ and $B$. In particular, when $k = k'$, the mixed derivative between $B_{ik}$ and $A_{kj}$ contains the additional term $(\mathbf{G}_W)_{ij}$, which arises directly from the bilinear parameterization $BA$. Whenever the right-hand side of Eq. (9) is nonzero for some $(i, j, k, k')$, the LoRA-parameter Hessian has a nonzero cross-block entry that cannot be represented by a diagonal preconditioner.*

Theorem 3.2 shows that coupling is not an artifact of optimization noise: it is *structurally induced* by the bilinear parameterization $BA$ and manifests as genuine cross-block curvature between the LoRA factors $A$ and $B$. The full proof is provided in Appendix E.2.

Geometrically, a diagonal preconditioner can only *rescale* coordinate axes; it cannot represent the *rotation/shear* encoded by off-diagonal curvature. Concretely, even in 2D, if $\mathbf{H} = \begin{bmatrix} 1 & \rho \\ \rho & 1 \end{bmatrix}$ with $\rho \neq 0$, then for any diagonal

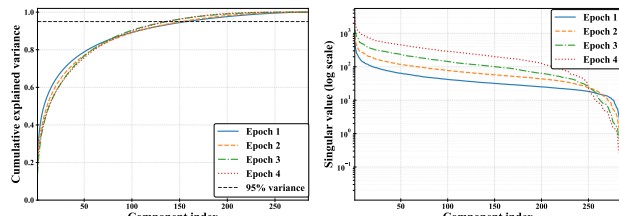

*(a)* Cumulative explained variance. *(b)* Singular value spectrum.

*Figure 1.* Spectral analysis of the MMLU validation gradient $\mathbf{G}_{\text{val}}$ on Llama2-7B. We decompose the gradient matrix via SVD to obtain singular values $\sigma_i$. **(a)** Cumulative explained variance. A steeper curve indicates that a **smaller principal subspace dimension** is sufficient to capture the majority of the variance (e.g., Rank 150 captures 95%), confirming high directional information density. **(b)** The singular values ($\sigma_k$) exhibit precipitous decay, further verifying the intrinsic low-rank structure.

$$\mathbf{D} = \text{diag}(d_1, d_2),$$

$$\|\mathbf{H} - \mathbf{D}\|_F^2 = (1 - d_1)^2 + (1 - d_2)^2 + 2\rho^2 \geq 2\rho^2, \tag{10}$$

so a diagonal surrogate suffers an irreducible error floor whenever curvature is rotated/coupled.

## 3.3. The Solution: Optimal Geometry Recovery via Spectral Filtering

Section 3.2 suggests that diagonal surrogates cannot represent the rotated, coupled geometry that governs targeted improvement. Thus, rather than refining a diagonal scaling, we construct a tractable *non-diagonal* operator that captures cross-parameter couplings while remaining computable at LLM scale. In the ideal influence proxy (Eq. (7)), the metric $\mathbf{H}_{\text{val},t}^\dagger$ measures alignment between the target gradient and a candidate update, but forming $\mathbf{H}_{\text{val},t}$ (let alone $\mathbf{H}_{\text{val},t}^\dagger$) is infeasible. Crucially, for ranking data we do not need an entrywise approximation of $\mathbf{H}_{\text{val},t}$; it suffices to recover a low-dimensional *coupled subspace* that concentrates the dominant task-relevant directions, which diagonal methods cannot express by construction.

Our construction starts from per-example validation gradients, which are already required to define the targeted direction. Let $\mathbf{G}_{\text{val},t} \in \mathbb{R}^{d \times |\mathcal{D}_{\text{val}}|}$ be the stacked gradient matrix whose columns are $\nabla_{\boldsymbol{\theta}} \ell(\boldsymbol{z}_{\text{val}}, \boldsymbol{\theta}_t)$ for $\boldsymbol{z}_{\text{val}} \in \mathcal{D}_{\text{val}}$. Define the gradient covariance proxy

$$\widehat{\mathbf{F}}_{\text{val},t} \triangleq \mathbf{G}_{\text{val},t} \mathbf{G}_{\text{val},t}^\top \in \mathbb{R}^{d \times d}, \tag{11}$$

which is *positive semidefinite (PSD)* and *non-diagonal*.

**Theorem 3.3** (Eigenspace stability of the proxy). *Let $\mathcal{S}_r(\mathbf{M})$ denote the top-$r$ eigenspace of a PSD matrix $\mathbf{M}$, and let $\Theta(\cdot, \cdot)$ denote the principal-angle matrix between two subspaces. For negative log-likelihood (NLL) objectives, the per-example Hessian admits the classical Gauss–Newton/Fisher decomposition, and there exists a quantity*

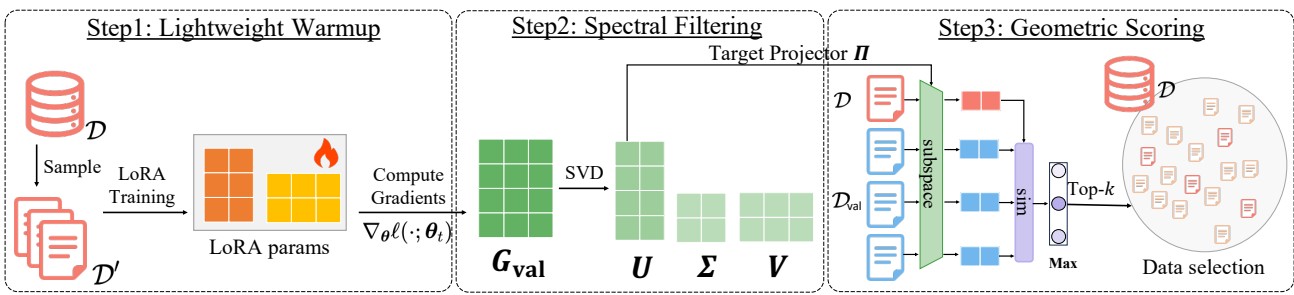

*Figure 2.* **Overview of GIST. Step 1: Lightweight warmup** performs a short LoRA warmup on a sampled subset and computes validation gradients. **Step 2: Spectral filtering** applies an SVD on the validation gradient matrix to construct a low-rank target subspace (Target projector). **Step 3: Geometric scoring** projects candidate gradients onto the target subspace and selects Top-$k$ samples.

$\varepsilon_t \geq 0$ *that summarizes (i) the residual-curvature magnitude and (ii) the proxy mismatch, such that*

$$\left\| \sin \Theta \big( \mathcal{S}_r(\mathbf{H}_{\text{val},t}), \ \mathcal{S}_r(\widehat{\mathbf{F}}_{\text{val},t}) \big) \right\|_2 \ \leq \ C \, \varepsilon_t, \qquad (12)$$

*where the constant $C$ depends only on the leading eigengap of the validation curvature.*

A full proof and formal conditions are provided in Appendix E.3. Theorem 3.3 implies that exact entrywise matching is unnecessary; instead, **it suffices for the loss to decrease to a low magnitude** to bound the subspace approximation error. This insight elucidates the requirement of a warm-up phase: it is required to steer the optimization trajectory out of the initial high-noise regime into a local basin where the curvature stabilizes. Crucially, since pre-trained models facilitate a rapid initial descent, this warm-up can be **extremely brief**. Empirical validation of these dynamics is provided in Appendix I.2.

We therefore recover the task subspace directly from $\mathbf{G}_{\text{val},t}$ via a compact SVD:

$$\mathbf{G}_{\text{val},t} = \mathbf{U}_t \boldsymbol{\Sigma}_t \mathbf{V}_t^\top, \qquad (13)$$

and define the rank-$r$ projector $\mathbf{P}_r \triangleq \mathbf{U}_{t,r} \mathbf{U}_{t,r}^\top$ using the top-$r$ left singular vectors. By Eckart–Young–Mirsky (Eckart & Young, 1936), $\mathbf{P}_r$ is the optimal rank-$r$ linear operator for capturing the dominant parameter-space variation of $\mathbf{G}_{\text{val},t}$ under squared reconstruction error. Importantly, $\mathbf{P}_r$ is explicitly non-diagonal: it represents a rotation in parameter space and can encode the coupled directions that diagonal preconditioners provably miss, yielding a tractable surrogate geometry for comparing candidate updates in Eq. (7).

# 4. GIST: The Proposed Methodology

Based on the theoretical analysis in Section 3, we propose GIST, a data selection framework that leverages spectral subspace filtering to robustly identify high-value training samples. GIST operates in three main steps: (1) capturing gradient trajectories via lightweight LoRA warmup, (2)

extracting the intrinsic target geometry via SVD, and (3) selecting data based on projected alignment. Full pipeline is illustrated in Fig. 2.

## 4.1. Subspace Gradient Feature Computation

**Step 1: Trajectory Collection via Lightweight Warmup.** To efficiently approximate the evolution of the optimization landscape without incurring the cost of full training, we perform a lightweight warmup. We randomly sample a small subset of the candidate pool (e.g., 5%), denoted as $\mathcal{D}' \subset \mathcal{D}$, and fine-tune the base LLM on $\mathcal{D}'$ using LoRA adapters (Hu et al., 2022) for an epoch. We collect the checkpoint $\boldsymbol{\theta}_t$. Let $d$ denote the number of trainable LoRA parameters. For each target example $\boldsymbol{z}_{\text{val}} \in \mathcal{D}_{\text{val}}$ and candidate example $\boldsymbol{z} \in \mathcal{D}$, we compute their gradients $\nabla_{\boldsymbol{\theta}} \ell(\cdot; \boldsymbol{\theta}_t) \in \mathbb{R}^d$ in the LoRA parameter space.

**Step 2: Extracting the Task Subspace.** Construct the target gradient matrix $\mathbf{G}_{\text{val},t} \in \mathbb{R}^{d \times |\mathcal{D}_{\text{val}}|}$. Then we perform Singular Value Decomposition (SVD) on $\mathbf{G}_{\text{val},t}$:

$$\mathbf{G}_{\text{val},t} = \mathbf{U}_t \boldsymbol{\Sigma}_t \mathbf{V}_t^\top. \qquad (14)$$

We determine the effective rank $r_t$ by analyzing the singular value spectrum (e.g., capturing 95% explained variance). The **Target Projector** is then defined by the top-$r$ singular vectors:

$$\boldsymbol{\Pi} \triangleq \mathbf{U}_r^\top \in \mathbb{R}^{r \times d}. \qquad (15)$$

This operator captures the principal directions of the task while discarding the orthogonal noise.

**Step 3: Geometric Scoring via Projected Alignment.** Let $\mathbf{g}_{\text{val},t}^{(j)} \triangleq \nabla_{\boldsymbol{\theta}} \ell(\boldsymbol{z}_{\text{val}}^{(j)}; \boldsymbol{\theta}_t) \in \mathbb{R}^d$. With the target projector $\boldsymbol{\Pi}$, we map gradients into the $r$-dimensional target subspace. For a candidate $\boldsymbol{z}_i$ and a specific target example $\boldsymbol{z}_{\text{val}}^{(j)} \in \mathcal{D}_{\text{val}}$, we define their **Subspace Alignment Score** at step $t$ as the cosine similarity of their projected gradients:

$$\text{Sim}_t(\boldsymbol{z}_i, \boldsymbol{z}_{\text{val}}^{(j)}) \triangleq \frac{(\boldsymbol{\Pi} \mathbf{g}_{i,t})^\top (\boldsymbol{\Pi} \mathbf{g}_{\text{val},t}^{(j)})}{\|\boldsymbol{\Pi} \mathbf{g}_{i,t}\|_2 \, \|\boldsymbol{\Pi} \mathbf{g}_{\text{val},t}^{(j)}\|_2}, \qquad (16)$$

Note that $\mathbf{g}_i = \nabla_{\boldsymbol{\theta}}\ell(\boldsymbol{z}_i;\boldsymbol{\theta}_t)$ aggregates token-level contributions (average over tokens), so its norm can vary substantially with sequence length and local loss scale. To prevent such magnitude effects from dominating the ranking, we use cosine similarity after projecting to the learned task subspace: normalization removes per-example scale variations, while $\boldsymbol{\Pi}$ retains only the coupled directions that are reproducible on $\mathcal{D}_{\text{val}}$. As a result, GIST scores candidates by directional agreement with the target geometry, rather than by raw gradient inner product that can be confounded by token aggregation.

### 4.2. Multi-Task Aggregation and Selection

In standard setting, the target set $\mathcal{D}_{\text{val}} = \{\boldsymbol{z}_{\text{val}}^{(1)}, \dots, \boldsymbol{z}_{\text{val}}^{(M)}\}$ is constructed such that each example represents a distinct task or capability. A candidate data point might be highly relevant to one specific task (e.g., math) but orthogonal to others (e.g., coding). Averaging the target gradients would dilute these specific gradient directions. Instead, we employ a **Maximum Relevance** strategy, following Xia et al. (2024); Ivison et al. (2025). For each candidate $\boldsymbol{z}_i$, we compute its alignment with every target example $j \in \{1, \dots, M\}$ using Eq. (16) and retain the maximum score:

$$\text{FinalScore}(\boldsymbol{z}_i) = \max_{j \in \{1, \dots, M\}} \text{Sim}_t(\boldsymbol{z}_i, \boldsymbol{z}_{\text{val}}^{(j)}). \qquad (17)$$

This strategy prioritizes *specialist* candidates that effectively support at least one target requirement, regardless of their utility for others. Finally, we select the top-$k$ candidates with the highest FinalScore for fine-tuning.

## 5. Experiments

### 5.1. Experimental Setup

**Training Datasets.** We adopt the settings from Xia et al. (2024) and utilize a heterogeneous pool of instruction tuning data, comprising approximately 270K examples. The pool integrates (1) task-specific datasets aggregated from sources like FLAN V2 (Longpre et al., 2023) and COT (Wei et al., 2022), with (2) open-ended, human-authored generation datasets such as DOLLY (Conover et al., 2023) and OPEN ASSISTANT 1 (Köpf et al., 2023). These sources exhibit significant variance in both instruction format and underlying reasoning logic. We note that this pool is designed to be disjoint from the target domain, ensuring that the selection process is tested in a rigorous transfer setting. See Appendix G.1 for additional details.
**Evaluation datasets.** Following the evaluation protocol established by Xia et al. (2024), we benchmark our method on three diverse datasets: MMLU (Hendrycks et al., 2020), TYDIQA (Clark et al., 2020), and BBH (Suzgun et al., 2023). MMLU serves as a broad knowledge test, covering 57 subjects ranging from elementary mathematics to law. TYDIQA

*Table 1.* Statistics of evaluation datasets. We select tasks covering diverse output formats, including multiple-choice, extractive spans, and generative reasoning.

| | | | Data Size | | |
| --- | --- | --- | --- | --- | --- |
| **Dataset** | **Shots** | **Tasks** | **Val.** | **Test** | **Output Format** |
| MMLU | 5 | 57 | 285 | 18,721 | Multiple Choice |
| TYDIQA | 1 | 9 | 9 | 1,713 | Extractive Span |
| BBH | 3 | 23 | 81 | 920 | Generation (CoT) |

evaluates multilingual reading comprehension across 9 typologically diverse languages, requiring the model to extract exact answers from passages. BBH is a curated suite of 27 challenging tasks from BIG-Bench, designed to probe complex reasoning capabilities. Table 1 provides detailed statistics. Consistent with Xia et al. (2024), we utilize the few-shot examples provided in each subtask for a dual purpose: they serve as the target set $\mathcal{D}_{\text{val}}$ to guide our data selection and subsequently act as in-context demonstrations during evaluation. See Appendix H for further details.

**Models for data selection and training.** We evaluate GIST with three representative base models: Llama2-7B (Touvron et al., 2023), Llama3.2-3B (Dubey et al., 2024), and Qwen2.5-1.5B (Team, 2024). Following standard efficient fine-tuning protocols, all training stages are conducted using LoRA (Hu et al., 2022). We report the average performance and standard deviation across three random seeds. Appendix G.3 provides further implementation details.

### 5.2. Baselines

We compare GIST against methods spanning four categories. First, as a naive baseline, we employ **Random Selection**. Second, we evaluate heuristics prioritizing intrinsic difficulty, including **Length** (Zhao et al., 2024a) and **PPL.** (Yin & Rush, 2025; Antonello et al., 2021), where we select examples with the *highest* perplexity. Third, we include similarity-based retrievers targeting semantic proximity to the validation set: standard **Embedding** similarity via GTR-Base (Ni et al., 2022) and **RDS+** (Ivison et al., 2025), which exploits the model's own hidden states. Finally, we compare against **LESS** (Xia et al., 2024), the state-of-the-art optimizer-based method selecting data via gradient projection and Adam-state rescaling. See Appendix F for details.

### 5.3. Main Results

Results are reported in Table 2. For LESS, following the official default setting, we run 4 epochs with random projection dimension 8192. For GIST, we run 1 warmup epoch, and use projection dimensions 150, 9, and 50 for MMLU, TYDIQA, and BBH, respectively. Gradient spectral analysis and rank selection are detailed in Appendix I.1.

**GIST matches or outperforms the state-of-the-art base-**

*Table 2.* Accuracy across datasets and models. *Base* denotes 0% (no selection) and *Full* denotes 100% (full dataset). All other methods select 5% of the data under the same finetuning budget. Gray ± values report standard deviations. **Bold** numbers denote the best selected subset in each row, and underlined numbers denote the second best selected subset. Avg. $\Delta$ reports the average absolute improvement over *Base* over MMLU, TYDIQA and BBH.

| Model | Dataset | Base (0%) | Full (100%) | Rand. | Length | PPL. | Embed. | RDS+ | LESS | GIST |
|-------|---------|-----------|-------------|-------|--------|------|--------|------|------|------|
| Llama2-7B | MMLU | *45.6* | *51.6* | 46.5±0.5 | 50.5±0.3 | 38.9±0.9 | 47.3±0.3 | 46.1±0.6 | 50.2±0.5 | **51.2**±0.7 |
| | TYDIQA | *46.4* | *54* | 52.7±0.4 | 51.1±0.3 | 32.9±0.4 | 49.8±0.8 | 51.0±0.3 | **56.2**±0.7 | 55.8±0.6 |
| | BBH | *38.3* | *43.2* | 38.9±0.5 | 39.8±0.7 | 34.5±0.2 | 38.6±0.6 | 39.0±0.7 | 41.5±0.6 | **41.8**±0.8 |
| | Avg. $\Delta$ | – | *+6.2* | +2.6 | +3.7 | -8.0 | +1.8 | +1.9 | +5.9 | **+6.2** |
| Llama3.2-3B | MMLU | *53.9* | *55.2* | 53.2±0.6 | **57.6**±0.9 | 51.8±0.3 | 53.9±0.6 | 51.9±0.2 | 56.5±0.6 | 56.1±0.4 |
| | TYDIQA | *60.4* | *66.6* | 64.1±0.4 | 63.9±1.3 | 57.1±0.7 | 62.8±0.4 | 62.6±0.5 | 67.1±0.8 | **69.2**±0.3 |
| | BBH | *45.5* | *47.8* | 45.1±0.2 | 45.3±0.3 | 43.0±0.4 | 44.6±0.5 | 45.1±0.4 | 46.1±0.7 | **48.0**±0.5 |
| | Avg. $\Delta$ | – | *+3.3* | +0.9 | +2.3 | -2.6 | +0.5 | -0.1 | +3.3 | **+4.5** |
| Qwen2.5-1.5B | MMLU | *62.0* | *62.3* | 61.5±0.3 | 62.8±0.1 | 61.7±0.2 | 62.3±0.2 | 62.4±0.2 | 62.2±0.2 | **62.9**±0.6 |
| | TYDIQA | *57.8* | *59.9* | 55.6±0.3 | 56.7±0.3 | 58.4±0.4 | 54.1±0.3 | 56.2±0.3 | **61.4**±0.8 | 61.2±0.5 |
| | BBH | *43.8* | *44.0* | 43.0±0.4 | 42.8±0.2 | 43.1±0.2 | 43.6±0.2 | 39.0±0.6 | 43.2±0.5 | **44.1**±0.4 |
| | Avg. $\Delta$ | – | *+0.9* | -1.2 | -0.4 | -0.1 | -1.2 | -2 | +1.1 | **+1.5** |

**line across diverse model architectures.** As shown in Table 2, GIST achieves the greatest average improvement (Avg. $\Delta$) across all three models, outperforming heuristic baselines and the previous SOTA method, LESS. On Llama2-7B, GIST achieves a +6.2 average improvement, matching the Full fine-tuning upper bound. The advantage is still pronounced on the more capable Llama3.2-3B and Qwen2.5-1.5B models, where GIST delivers improvements of +4.5 and +1.5, respectively, surpassing LESS (which achieves +3.3 and +1.1). This demonstrates that GIST's geometric subspace approach can scale effectively to modern, stronger base models.

**GIST mitigates negative transfer by surpassing fine-tuning on the full dataset**. A remarkable finding is that GIST, using only 5% of the data, frequently matches or exceeds the performance of fine-tuning on the full dataset (100%). Specifically, on Llama3.2-3B, GIST outperforms Full fine-tuning by a substantial margin (+4.5 vs. +3.3), and on Qwen2.5-1.5B, it achieves nearly double the gain of the full dataset (+1.5 vs. +0.9). This indicates that the full dataset likely contains redundant or noisy samples that hinder optimization, whereas GIST isolates task-relevant update directions, allowing the model to learn more effectively.

**GIST demonstrates superior robustness compared to heuristic and optimizer-based baselines.** While heuristics like *Length* occasionally perform well on specific metrics (e.g., MMLU on Llama3.2), they lack consistency, often degrading performance on other tasks (e.g., negative average gains on Qwen). Similarly, while LESS generally performs well, it struggles on Qwen2.5-1.5B, degrading on BBH (43.2 vs. Base 43.8). In contrast, GIST maintains significant positive gains across all tasks and models, proving that subspace alignment is a more robust proxy for Eq. (7) than curvature-based influence or simple surface statistics.

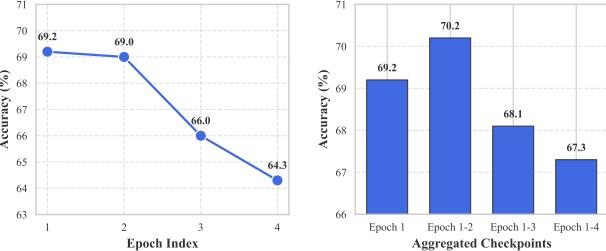

*(a)* Single Epoch Performance   *(b)* Cumulative Performance
*Figure 3.* **Impact of Checkpoint Selection.** (a) Using single-epoch gradients shows a clear performance drop in later epochs. (b) Aggregating multiple checkpoints (weighted) does not outperform the early-stop strategy, confirming that early gradients contain the essential task optimization directions.

*Table 3.* Number of checkpoints ($N$) used for select data with GIST for Llama3.2-3B.

| | MMLU | TYDIQA | BBH | Avg. |
|---|------|--------|-----|------|
| LESS | 56.5±0.6 | 67.1±0.8 | 46.1±0.7 | 56.6 |
| $N = 4$ | 56.3±0.3 | 67.3±0.1 | 46.8±0.4 | 56.8 |
| $N = 1$ (default) | 56.1±0.4 | 69.2±0.3 | 48.0±0.2 | **57.8** |

**5.4. Sensitivity**

**Using only early checkpoints is sufficient; later checkpoints can even hurt performance.** Our findings offer a nuanced perspective on checkpoint aggregation compared to Xia et al. (2024). While aggregating influence scores can stabilize the zig-zagging Adam trajectory near convergence (Figure 5b) and partly mitigate the limited expressivity of a single-checkpoint diagonal surrogate, we find that for geometric subspace alignment, **the most valuable directional information is intrinsically concentrated in the early training stage**. As illustrated in Figure 3a, the performance of data selection using single-epoch gradients shows a monotonic decline on TYDIQA with Llama3.2-3B, dropping from 69.2% (Epoch 1) to 64.3% (Epoch 4).

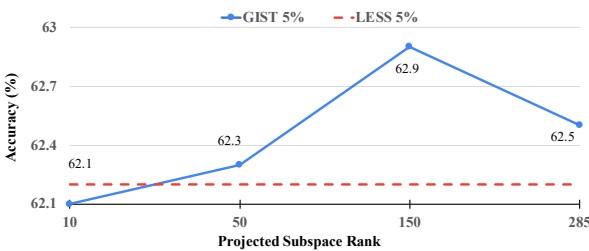

*Figure 4.* Accuracy as a function of the projection rank used by GIST, compared with LESS at the same selection budget.

Even when aggregating checkpoints using the weighted scheme proposed in LESS (Figure 3b), adding later checkpoints yields diminishing returns and eventually harms performance: while combining Epochs 1-2 yields a slight peak (70.2%), incorporating all four epochs degrades accuracy to 67.3%, significantly lower than using the early checkpoints alone. This aligns with the results in Table 3. We attribute this phenomenon to the evolution of the gradient geometry shown in Figure 1. As training progresses, the spectral distribution of the target gradient matrix becomes increasingly concentrated on the leading singular values. This shrinkage in intrinsic dimension suggests that late-stage gradients may lose the diverse directional information present in early checkpoints, discarding critical optimization directions required for robust data selection.

**Spectral filtering is essential, yet identifying the dominant optimization direction is the primary driver of performance.** Figure 4 presents the sensitivity analysis of GIST regarding subspace rank $r$ on MMLU ($|\mathcal{D}_{\text{val}}| = 285$). We observe that performance does not increase monotonically with rank; instead, it peaks at $r = 150$ (62.9%) and drops to 62.5% using full rank ($r = 285$). This confirms the necessity of spectral filtering to exclude noisy tail components that may harm generalization. Surprisingly, even at an extremely low rank $r = 10$ which captures only 40% of the cumulative explained variance, GIST achieves an accuracy of 62.1%, comparable to LESS. This further validates our core idea: effective data selection depends on **maintaining directional consistency with task subspace** instead of preserving the exact inner products of gradients. We include a detailed subspace analysis in Appendix J.

### 5.5. Efficiency

**Compared to LESS, GIST is highly efficient in practice, achieving better performance with only about one-quarter of LESS's wall-clock runtime for a single task.** Table 4 summarizes the asymptotic complexity and wall-clock runtime of each stage in GIST compared with LESS. All wall-clock time is measured in **single A100 (80GB) GPU hours**. Similar to LESS, the dominant cost comes from computing per-example gradient features, which scales linearly with the candidate dataset size

*Table 4.* Efficiency comparison between GIST (Ours) and LESS. Runtime is measured in **single A100 GPU hours**. GIST significantly reduces the overhead in Warmup and Feature Extraction stages (approx. $4\times$ speedup) and requires negligible time for SVD.

| Method | Metric | Warmup | Target SVD | Grad. Feats. |
|---|---|---|---|---|
| **LESS** | Time | 6.0 h | – | 48.0 h |
| | Compl. | $\mathcal{O}(|\mathcal{D}'| \cdot N)$ | – | $\mathcal{O}(|\mathcal{D}| \cdot N)$ |
| **GIST** | Time | **1.5 h** | **< 1 m** | **12.0 h** |
| | Compl. | $\mathcal{O}(|\mathcal{D}'|)$ | $\mathcal{O}(|\mathcal{D}_{\text{val}}|^2 d)$ | $\mathcal{O}(|\mathcal{D}|)$ |

$|\mathcal{D}|$ and the per-step training cost. In practice, GIST is substantially cheaper overall because it requires only *one* epoch (i.e., a single checkpoint) to form the target subspace, whereas LESS typically aggregates gradients across multiple epochs/checkpoints; this yields an end-to-end runtime that is roughly $\sim 4\times$ smaller under the same setup.

**The target SVD is cheap to compute in GIST.** A key additional cost unique to GIST is the *target SVD* used to construct the low-rank projector. Importantly, this step is inexpensive in both theory and practice. Concretely, we avoid forming any $d \times d$ matrix and instead compute the sample-space Gram matrix $\mathbf{G}_{\text{val},t}^{\top}\mathbf{G}_{\text{val},t}$ on the target set, followed by an eigendecomposition on an $|\mathcal{D}_{\text{val}}| \times |\mathcal{D}_{\text{val}}|$ matrix (with chunked multiplication), so the compute primarily scales with the small target size $|\mathcal{D}_{\text{val}}|$ and the chosen rank $r$. Empirically, constructing the projection is fast: for TYDIQA with $|\mathcal{D}_{\text{val}}| = 9$, it takes only 3.51s on average; for BBH ($|\mathcal{D}_{\text{val}}| = 69$) and MMLU ($|\mathcal{D}_{\text{val}}| = 285$), the average time is 22.54s and 61.06s, respectively. This confirms that the target-SVD overhead is negligible compared to gradient feature computation, and the overall runtime improvement of GIST mainly comes from using a single-epoch gradient store.

**GIST is extremely storage-efficient in practice.** For example, when storing gradient features for Qwen2.5-1.5B over the three tasks, GIST uses only 217MB on disk, while LESS requires 75GB under the same setting, i.e., $\approx 350\times$ larger storage (about 99.7% more disk usage). We follow the standard LESS setup with four training datasets totaling $\sim 270$k examples and a projection dimension of 8192. In contrast to LESS, which applies a Johnson–Lindenstrauss random projection (Johnson et al., 1984) with entries drawn from a Rademacher distribution and stores the resulting projected gradient features across multiple checkpoints, GIST performs a lightweight target SVD to recover a very low-dimensional task-relevant subspace and then projects gradients onto this target subspace. Together with single-epoch extraction, this leads to a dramatically smaller disk footprint.

## 6. Conclusion

In this work, we presented a unified optimization perspective on data selection, revealing that prior methods funda-

mentally suffer from geometric misalignment—either by ignoring curvature or relying on restrictive diagonal approximations. We demonstrated that in PEFT, the optimization landscape exhibits a rotated, low-rank structure that diagonal preconditioners fail to capture. To address this, we proposed `GIST`, which leverages spectral filtering to recover the coupled task geometry. `GIST` achieves state-of-the-art performance with significantly reduced computational overhead, validating that correctly modeling optimization geometry, rather than merely scaling up selection complexity, is key to efficient targeted instruction tuning.

## Acknowledgments

The authors would like to thank the anonymous reviewers for their constructive comments. This work was supported in part by the Commonwealth Cyber Initiative (CCI) under Award No. VV-1Q26-005 and the National Science Foundation under Grant No. 2331315. Any opinions, findings, and conclusions or recommendations expressed in this material are those of the authors and do not necessarily reflect the views of the National Science Foundation.

## Impact Statement

This paper studies influence-based data selection for targeted instruction tuning of large language models, with the goal of improving data efficiency and training effectiveness. The primary intended benefits are reduced compute and cost for fine-tuning, faster iteration, and better performance on specified target tasks or domains.

Our methods may also introduce or amplify risks. Because influence scores prioritize examples that most affect a chosen target set, they can reinforce target-set biases, overfit to narrow benchmarks, or de-emphasize underrepresented groups and topics if the target distribution is skewed. In addition, data selection can inadvertently increase the presence of toxic, copyrighted, private, or otherwise sensitive content if such examples are deemed influential for the target objective. These concerns are not unique to our approach but are particularly relevant when automating dataset curation.

To mitigate these risks, we recommend pairing influence-based selection with standard data governance practices, including privacy and licensing checks, toxicity and safety filtering, and audits of demographic and topical coverage. We also encourage reporting target-set construction, selection criteria, and downstream evaluation beyond the target benchmarks (e.g., robustness and safety) to reduce the chance of unintended harms.

Overall, this work aims to advance machine learning methodology, and its broader impacts will depend on how the selected data and resulting models are deployed and governed.

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

# A. Limitations and Future Work

While GIST offers a principled geometric view, our current instantiation is intentionally simple—a minimal, proof-driven realization of the theory. Empirically, we find that applying GIST early in training (after a lightweight warmup) is often most beneficial; in our experiments, using it within the first two epochs can yield stronger performance. Developing more expressive variants and adaptive mechanisms that automatically decide *when* and *how* to apply GIST remains an important direction.

**Granularity and Magnitude.** First, following standard practices in efficient data selection (Xia et al., 2024; Nikdan et al., 2025), we utilize sequence-level gradients and cosine similarity. While computationally robust, this choice emphasizes *directional alignment* and de-emphasizes gradient magnitude (a proxy for difficulty) as well as token-level granularity. As a result, GIST may under-separate samples whose gradients share similar directions but differ substantially in scale, and token-level signals are averaged out within each sequence representation.

**Warmup Requirement.** Second, our approximation relies on the Gauss–Newton decomposition (Lemma E.2), which is most accurate once optimization enters a relatively stable regime. We satisfy this with a short warmup phase. While our experiments suggest the relevant subspace stabilizes quickly, our current analysis does not yet characterize the transient dynamics in the very early optimization steps *before* warmup, which could potentially be leveraged for further gains.

**Scale vs. Direction.** Third, unlike optimizer-based approaches that implicitly capture curvature scale (e.g., Adam's second moment), GIST focuses on recovering *principal directions* (an eigenspace). This is a deliberate design to avoid numerical issues associated with ill-conditioned curvature in LLMs. However, how to incorporate reliable *scaling* for the recovered directions (e.g., via lightweight diagonal or low-rank scale estimates) remains open.

**Future Directions.** Motivated by the above trade-offs, future work includes: (1) improving the fidelity of the recovered subspace (e.g., reducing information loss introduced by spectral filtering), and (2) developing principled and practical estimators for the effective dimensionality and stage-dependent geometry of the optimization manifold. A deeper understanding of what dominant optimization directions represent may further inform data selection and related downstream interventions grounded in optimization geometry.

# B. Extended Discussion on Related Work

**Instruction Data Selection.** The transition from data quantity to quality has become a central theme in LLM training. Early work focused on heuristic filtering based on surface-level metrics like perplexity or instruction length (Zhou et al., 2023; Chen et al., 2024; Li et al., 2024b). While efficient, these methods are often *task-agnostic*, ignoring the specific requirements of the target distribution. To address this, similarity-based approaches utilize embedding retrieval to select data semantically close to the target (Ni et al., 2022; Liu et al., 2024). However, semantic similarity in the pre-trained embedding space does not necessarily translate to gradient alignment during fine-tuning. Our work targets the more rigorous *gradient-based* setting, aiming to select data that explicitly optimizes the training objective on a target domain.

**Scalable Influence and Data Attribution Approximations.** Influence functions (Koh & Liang, 2017) provide a principled framework for data attribution and selection by estimating the effect of a training sample on validation loss, but their classical form requires Hessian-inverse computations that are infeasible for billion-scale models. Recent works therefore develop scalable approximations, including TRAK (Park et al., 2023), DataInf (Kwon et al., 2024), scalable fine-tuning from multiple data sources (Li et al., 2024a), influence distillation (Nikdan et al., 2025), and influence-preserving proxy models for gradient-based data selection (Chen et al., 2026). Closest to our setting, LESS (Xia et al., 2024) selects instruction-tuning data by comparing projected gradients and using optimizer statistics such as Adam's second moment to approximate influence. While these methods make attribution practical at LLM scale, they mainly focus on scalable gradient matching or optimizer-dependent approximations. In contrast, GIST directly recovers a low-rank target subspace from validation gradients and scores candidates by projected directional alignment, replacing diagonal optimizer heuristics with task-specific geometric subspace recovery.

**Spectral Properties of Deep Optimization.** Our approach is grounded in the spectral analysis of neural network optimization. It is well-established that the Hessian and gradient covariance of deep networks exhibit a "bulk-and-outlier" or heavy-tailed spectral structure (Sagun et al., 2016; Papyan, 2020; Ghorbani et al., 2019): the optimization trajectory is often concentrated in a low-dimensional manifold, while the remaining directions are dominated by stochastic noise (Li et al., 2018a;b). This view has also motivated recent LLM optimizers that move beyond purely coordinate-wise updates

by exploiting spectral or rotation-aware structure, such as GaLore (Zhao et al., 2024b), SOAP (Vyas et al., 2025), and Muon (Jordan et al., 2024). These methods show that identifying and updating within well-conditioned low-dimensional or rotated subspaces can improve large-scale optimization. `GIST` applies a similar geometric principle to gradient-based data selection: instead of unstable curvature inversion or diagonal optimizer-state rescaling, we perform spectral filtering on validation gradients and score candidates by their alignment with the resulting task subspace. This connects data selection with the intrinsic dimension of the optimization trajectory, ensuring that selection is driven by stable task-relevant directions rather than noisy or axis-aligned artifacts.

## C. Notations and Symbols

We summarize the mathematical notations used throughout the paper in Table 5.

*Table 5.* Summary of notations used in this paper.

| Symbol | Description |
| --- | --- |
| | *Datasets and Models* |
| $\mathcal{D}$ | Large-scale instruction tuning candidate pool, $\mathcal{D} = \{z_i\}_{i=1}^{|\mathcal{D}|}$ |
| $z$ | A data instance $z = (x, y)$ |
| $\mathcal{D}_{\mathrm{val}}$ | Target validation set defining the target distribution/task |
| $\mathcal{D}_{\mathrm{test}}$ | Held-out test set for evaluation |
| $S$ | Selected subset from the candidate pool, $S \subseteq \mathcal{D}$ |
| $k$ | Selection budget (cardinality constraint), $|S| = k$ |
| $J$ | Number of target tasks/requirements represented in $\mathcal{D}_{\mathrm{val}}$ |
| $z_{\mathrm{val}}^{(j)}$ | The $j$-th target/validation example (task requirement), $j \in \{1, \dots, J\}$ |
| $\mathcal{M}_{\boldsymbol{\theta}}$ | LLM with trainable parameters $\boldsymbol{\theta} \in \mathbb{R}^d$ (e.g., LoRA parameters) |
| $d$ | Number of trainable parameters (e.g., LoRA parameters) |
| $\boldsymbol{\theta}_t$ | Trainable parameters at checkpoint/step $t$ |
| $N$ | Number of collected checkpoints along warmup trajectory |
| | *Losses, Gradients, and Curvature* |
| $\ell(z, \boldsymbol{\theta})$ | Sample-wise loss at parameters $\boldsymbol{\theta}$ |
| $\mathcal{L}(S, \boldsymbol{\theta})$ | Empirical risk over a dataset $S$: $\mathcal{L}(S, \boldsymbol{\theta}) = \frac{1}{|S|} \sum_{z \in S} \ell(z, \boldsymbol{\theta})$ |
| $\mathbf{g}_t$ | Instantaneous (per-sample) training gradient at step $t$: $\mathbf{g}_t = \nabla_{\boldsymbol{\theta}} \ell(z_t, \boldsymbol{\theta}_t)$ |
| $\mathbf{H}_t$ | Instantaneous (per-sample) Hessian at step $t$: $\mathbf{H}_t = \nabla_{\boldsymbol{\theta}}^2 \ell(z_t, \boldsymbol{\theta}_t)$ |
| $\mathbf{H}_{\mathrm{val},t}$ | Validation Hessian: $\mathbf{H}_{\mathrm{val},t} = \nabla_{\boldsymbol{\theta}}^2 \mathcal{L}(\mathcal{D}_{\mathrm{val}}, \boldsymbol{\theta}_t)$ |
| $\mathbf{H}_{\mathrm{val},t}^{\dagger}$ | Moore–Penrose pseudoinverse of $\mathbf{H}_{\mathrm{val},t}$ |
| $\eta$ | Learning rate (step size) |
| $\epsilon$ | Numerical stability constant in Adam update |
| | *Adam Optimizer Statistics* |
| $\mathbf{m}_t$ | First-moment estimate (EMA of gradients) at step $t$ |
| $\mathbf{v}_t$ | Second-moment estimate (EMA of squared gradients) at step $t$ |
| $\hat{\mathbf{m}}_t$ | Bias-corrected first moment at step $t$ |
| $\hat{\mathbf{v}}_t$ | Bias-corrected second moment at step $t$ |
| | *Target Gradient Matrix and Spectral Subspace* |
| $\mathbf{g}_{i,t}$ | Per-sample gradient for candidate $z_i$ at checkpoint $t$: $\mathbf{g}_{i,t} = \nabla_{\boldsymbol{\theta}} \ell(z_i, \boldsymbol{\theta}_t) \in \mathbb{R}^d$ |
| $\mathbf{g}_{\mathrm{val},t}^{(j)}$ | Per-sample gradient for target example $z_{\mathrm{val}}^{(j)}$ at checkpoint $t$: $\mathbf{g}_{\mathrm{val},t}^{(j)} = \nabla_{\boldsymbol{\theta}} \ell(z_{\mathrm{val}}^{(j)}, \boldsymbol{\theta}_t) \in \mathbb{R}^d$ |
| $\mathbf{G}_{\mathrm{val},t}$ | Stacked target gradient matrix at checkpoint $t$: columns are $\mathbf{g}_{\mathrm{val},t}^{(j)}$, $\mathbf{G}_{\mathrm{val},t} \in \mathbb{R}^{d \times |\mathcal{D}_{\mathrm{val}}|}$ |
| $\mathbf{U}_t, \boldsymbol{\Sigma}_t, \mathbf{V}_t$ | Compact SVD: $\mathbf{G}_{\mathrm{val},t} = \mathbf{U}_t \boldsymbol{\Sigma}_t \mathbf{V}_t^{\top}$ |
| $r_t$ | Effective rank estimated from the spectrum at checkpoint $t$ |
| $r$ | Global subspace dimension used for scoring (e.g., $r = \max_t r_t$) |
| $\mathbf{U}_r$ | Top-$r$ left singular vectors of $\mathbf{G}_{\mathrm{val},t}$, $\mathbf{U}_r \in \mathbb{R}^{d \times r}$ |
| $\boldsymbol{\Pi}$ | Spectral projector to target subspace, $\boldsymbol{\Pi} = \mathbf{U}_r^{\top} \in \mathbb{R}^{r \times d}$ |
| $\mathbf{P}_r$ | Rank-$r$ orthogonal projector in parameter space, $\mathbf{P}_r = \mathbf{U}_r \mathbf{U}_r^{\top} \in \mathbb{R}^{d \times d}$ |

| Symbol | Description |
|---|---|
| | *Projected Alignment and Selection Scores* |
| $\mathrm{Sim}_t(\boldsymbol{z}_i, \boldsymbol{z}_{\mathrm{val}}^{(j)})$ | Subspace cosine similarity at checkpoint $t$ (Eq. (16)) |
| $\mathrm{FinalScore}(\boldsymbol{z}_i)$ | Aggregated score for selection, $\mathrm{FinalScore}(\boldsymbol{z}_i) = \max_{j \in \{1,\dots,J\}} \mathrm{Sim}_t(\boldsymbol{z}_i, \boldsymbol{z}_{\mathrm{val}}^{(j)})$ |

## D. Details of the Adam Optimizer and Gradient Rescaling

In this section, we provide the detailed formulation of the Adam optimizer (Kingma & Ba, 2015) used in fine-tuning Large Language Models (LLMs) and discuss how its second-moment statistics are utilized to approximate the optimization landscape for data selection methods like LESS (Xia et al., 2024).

### D.1. Standard Adam Update Rule

Let $\ell(\boldsymbol{z}, \boldsymbol{\theta})$ denote the loss function for a data sample $\boldsymbol{z}$ and model parameters $\boldsymbol{\theta} \in \mathbb{R}^d$. At each training step $t$, we compute the stochastic gradient on a mini-batch $\mathcal{B}_t$:

$$\mathbf{g}_t \triangleq \frac{1}{|\mathcal{B}_t|} \sum_{\boldsymbol{z} \in \mathcal{B}_t} \nabla_{\boldsymbol{\theta}} \ell(\boldsymbol{z}, \boldsymbol{\theta}_t). \tag{18}$$

Adam estimates the first moment (mean) $\mathbf{m}_t$ and the second raw moment (uncentered variance) $\mathbf{v}_t$ of the gradients using exponential moving averages (EMA):

$$\mathbf{m}_{t+1} = \beta_1 \mathbf{m}_t + (1 - \beta_1) \mathbf{g}_t, \tag{19}$$

$$\mathbf{v}_{t+1} = \beta_2 \mathbf{v}_t + (1 - \beta_2)(\mathbf{g}_t \odot \mathbf{g}_t), \tag{20}$$

where $\odot$ denotes the element-wise product, and $\beta_1, \beta_2 \in [0, 1)$ are the decay rates (typically $\beta_1 = 0.9, \beta_2 = 0.999$). To counteract the initialization bias (since $\mathbf{m}_0$ and $\mathbf{v}_0$ are initialized to zero vectors), bias-corrected estimates are computed as:

$$\hat{\mathbf{m}}_{t+1} = \frac{\mathbf{m}_{t+1}}{1 - \beta_1^{t+1}}, \quad \hat{\mathbf{v}}_{t+1} = \frac{\mathbf{v}_{t+1}}{1 - \beta_2^{t+1}}. \tag{21}$$

The parameters are then updated via:

$$\boldsymbol{\theta}_{t+1} = \boldsymbol{\theta}_t - \eta \cdot \frac{\hat{\mathbf{m}}_{t+1}}{\sqrt{\hat{\mathbf{v}}_{t+1}} + \epsilon}, \tag{22}$$

where $\eta$ is the learning rate and $\epsilon$ is a small constant for numerical stability. Structurally, the term $(\sqrt{\hat{\mathbf{v}}_{t+1}} + \epsilon)^{-1}$ acts as a **diagonal preconditioner**, rescaling the gradient based on the historical curvature of the optimization landscape.

### D.2. Toy Example: Diagonal vs. Full-Matrix Preconditioning under Coupling

To complement the Adam update in Eq. (22), we visualize a 2D quadratic objective under *non-coupled* (axis-aligned) versus *coupled* (rotated) curvature. Consider

$$L(\boldsymbol{\theta}) = \frac{1}{2} \boldsymbol{\theta}^{\top} \mathbf{H} \boldsymbol{\theta}, \qquad \nabla L(\boldsymbol{\theta}) = \mathbf{H} \boldsymbol{\theta}, \tag{23}$$

with the common initialization $\boldsymbol{\theta}_0 = (-2.5, 0)^{\top}$. This choice places the start on a coordinate axis so that, in the *non-coupled* case, the update direction does not introduce an artificial "orbit" caused by momentum; hence the visual difference is dominated by *coupling* rather than initialization.

**Two geometries (same spectrum, different orientation).** We compare an axis-aligned quadratic and its rotated counterpart:

$$\textbf{Non-coupled (axis-aligned):} \quad \mathbf{H}_{\mathrm{diag}} = \mathrm{diag}(20, 1), \tag{24}$$

$$\textbf{Coupled (rotated):} \quad \mathbf{H}_{\mathrm{cpl}} = \mathbf{R}\,\mathbf{H}_{\mathrm{diag}}\,\mathbf{R}^{\top}, \tag{25}$$

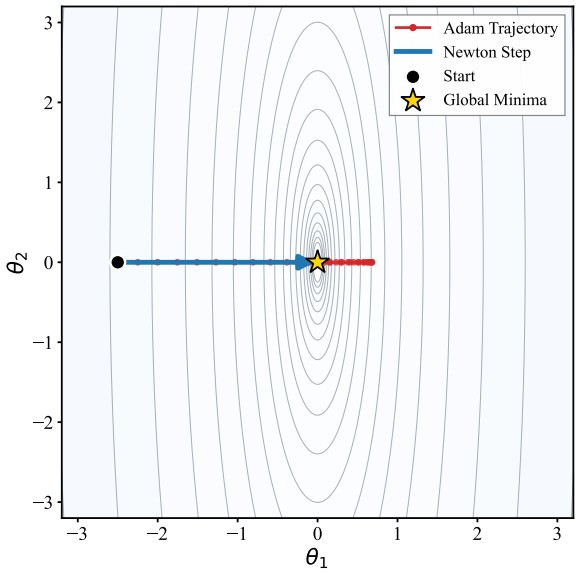

*(a)* Non-coupled (axis-aligned) quadratic: $\mathbf{H}_{\text{diag}} = \text{diag}(20, 1)$.

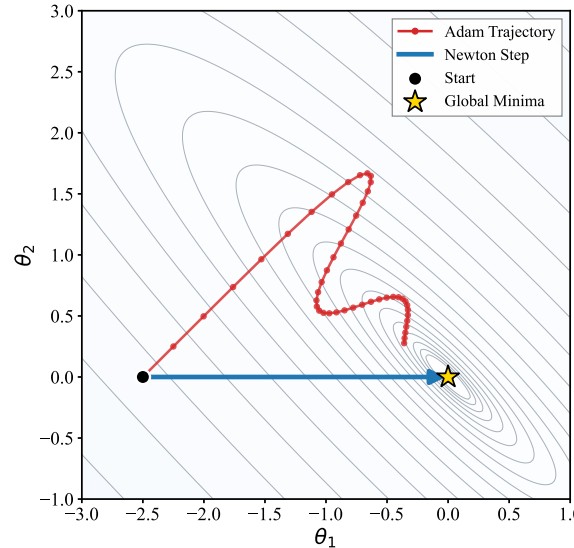

*(b)* Coupled (rotated) quadratic: $\mathbf{H}_{\text{cpl}}$ has large off-diagonal entries.

*Figure 5.* Toy 2D optimization dynamics with the same initialization $\boldsymbol{\theta}_0 = (-2.5, 0)$. Newton (full-matrix) follows the direct descent direction, while Adam (diagonal) cannot express the rotation induced by coupling, leading to a "zig-zag" trajectory on the coupled landscape.

where $\mathbf{R}$ is a 2D rotation matrix. Thus, both cases share the same eigenvalues (and condition number), but $\mathbf{H}_{\text{cpl}}$ is *not diagonal in the coordinate basis*, yielding tilted level sets due to off-diagonal coupling. For visualization we instantiate $\mathbf{H}_{\text{cpl}} = \begin{bmatrix} 10.5 & 9.5 \\ 9.5 & 10.5 \end{bmatrix}$.

**Newton step (full-matrix preconditioning).** Newton's method uses the full curvature:

$$\boldsymbol{\theta}_{t+1} = \boldsymbol{\theta}_t - \eta_{\text{Newton}} \mathbf{H}^{-1} \nabla L(\boldsymbol{\theta}_t). \tag{26}$$

With $\eta_{\text{Newton}} = 1$ and $\nabla L(\boldsymbol{\theta}_t) = \mathbf{H}\boldsymbol{\theta}_t$, Newton reaches the minimizer $\mathbf{0}$ in a single step on this quadratic, correcting both *scaling* and *rotation*.

**Adam (diagonal preconditioning).** We run standard Adam with $(\beta_1, \beta_2) = (0.9, 0.999)$ for $T = 45$ steps, using initial learning rate $\eta = 0.25$ with a linear decay schedule. Since Adam's preconditioner is diagonal, it can only rescale coordinates independently and cannot represent the off-diagonal coupling present in $\mathbf{H}_{\text{cpl}}$. As a result, on the coupled landscape, Adam may drift across the narrow valley and exhibit a less direct trajectory, whereas on the non-coupled landscape it behaves in a more stable, axis-consistent manner.

**Takeaway.** This toy example highlights the key geometric distinction: *full-matrix* curvature information can correct parameter coupling (rotation), whereas *diagonal* preconditioning cannot, which can lead to inefficient trajectories when the optimization landscape is strongly coupled.

## E. Theoretical Proof

### E.1. Proof of Theorem 3.1

**Theorem 3.1** (Single-level Approximate Optimization). *Fix the current parameters $\boldsymbol{\theta}_t$. Under the first-order approximation, Problem 1 is approximately reduced to maximizing the predicted reduction in validation loss:*

$$
\begin{aligned}
\max_{S \subseteq \mathcal{D}} \quad & \nabla_{\boldsymbol{\theta}} \mathcal{L}(\mathcal{D}_{val}, \boldsymbol{\theta}_t)^{\top} \mathbf{H}_{val,t}^{\dagger} \nabla_{\boldsymbol{\theta}} \mathcal{L}(S, \boldsymbol{\theta}_t) \\
\text{s.t.} \quad & |S| = k,
\end{aligned}
\tag{27}
$$

*up to constants independent of $S$ and higher-order terms.*

*Proof.* We write $\mathcal{L}_{\text{val}}(\boldsymbol{\theta}) := \mathcal{L}(\mathcal{D}_{\text{val}}, \boldsymbol{\theta})$. Let

$$\mathbf{g}_{\text{val},t} := \nabla_{\boldsymbol{\theta}} \mathcal{L}_{\text{val}}(\boldsymbol{\theta}_t), \qquad \mathbf{H}_{\text{val},t} := \nabla_{\boldsymbol{\theta}}^2 \mathcal{L}_{\text{val}}(\boldsymbol{\theta}_t), \tag{28}$$

and for any subset $S$,

$$\mathbf{g}_{S,t} := \nabla_{\boldsymbol{\theta}} \mathcal{L}(S, \boldsymbol{\theta}_t). \tag{29}$$

Problem 1 selects $S$ to minimize the validation loss after training on $S$, i.e., it compares $\mathcal{L}_{\text{val}}(\boldsymbol{\theta}_S^*)$ across different $S$, where $\boldsymbol{\theta}_S^*$ is the output of the inner optimization (exact minimizer or the terminal point of a training procedure) starting from $\boldsymbol{\theta}_t$.

First, we approximate the intractable mapping $S \mapsto \boldsymbol{\theta}_S^*$ around $\boldsymbol{\theta}_t$ by a single geometry-aware descent step using the validation curvature:

$$\widetilde{\boldsymbol{\theta}}_S \triangleq \boldsymbol{\theta}_t - \eta \, \mathbf{H}_{\text{val},t}^{\dagger} \, \mathbf{g}_{S,t}, \tag{30}$$

where $\eta > 0$ is a small stepsize and $\mathbf{H}_{\text{val},t}^{\dagger}$ denotes the Moore–Penrose pseudoinverse. This surrogate is a first-order approximation in the sense that it uses only first-order information of the subset loss at $\boldsymbol{\theta}_t$ and a fixed local metric at $\boldsymbol{\theta}_t$.

Then we assume $\mathcal{L}_{\text{val}}$ is twice continuously differentiable in a neighborhood of $\boldsymbol{\theta}_t$. Applying Taylor's theorem to $\mathcal{L}_{\text{val}}(\widetilde{\boldsymbol{\theta}}_S)$ around $\boldsymbol{\theta}_t$ yields

$$\mathcal{L}_{\text{val}}(\widetilde{\boldsymbol{\theta}}_S) = \mathcal{L}_{\text{val}}(\boldsymbol{\theta}_t) + \mathbf{g}_{\text{val},t}^{\top}(\widetilde{\boldsymbol{\theta}}_S - \boldsymbol{\theta}_t) + \frac{1}{2}(\widetilde{\boldsymbol{\theta}}_S - \boldsymbol{\theta}_t)^{\top} \mathbf{H}_{\text{val},t}(\widetilde{\boldsymbol{\theta}}_S - \boldsymbol{\theta}_t) + R_S, \tag{31}$$

where the remainder term satisfies $R_S = o(\|\widetilde{\boldsymbol{\theta}}_S - \boldsymbol{\theta}_t\|^2)$ as $\eta \to 0$.

Substituting (30) into the linear term in (31) gives

$$\mathbf{g}_{\text{val},t}^{\top}(\widetilde{\boldsymbol{\theta}}_S - \boldsymbol{\theta}_t) = -\eta \, \mathbf{g}_{\text{val},t}^{\top} \mathbf{H}_{\text{val},t}^{\dagger} \mathbf{g}_{S,t}. \tag{32}$$

The quadratic term in (31) scales as $\mathcal{O}(\eta^2)$:

$$\frac{1}{2}(\widetilde{\boldsymbol{\theta}}_S - \boldsymbol{\theta}_t)^{\top} \mathbf{H}_{\text{val},t}(\widetilde{\boldsymbol{\theta}}_S - \boldsymbol{\theta}_t) = \frac{\eta^2}{2} \mathbf{g}_{S,t}^{\top} \mathbf{H}_{\text{val},t}^{\dagger} \mathbf{H}_{\text{val},t} \mathbf{H}_{\text{val},t}^{\dagger} \mathbf{g}_{S,t}, \tag{33}$$

which is $\mathcal{O}(\eta^2)$ and hence is a higher-order term relative to the $\mathcal{O}(\eta)$ linear improvement term. (Here $\mathbf{H}^{\dagger} \mathbf{H} \mathbf{H}^{\dagger}$ is well-defined even when $\mathbf{H}$ is singular.)

Therefore, combining the above,

$$\mathcal{L}_{\text{val}}(\widetilde{\boldsymbol{\theta}}_S) = \mathcal{L}_{\text{val}}(\boldsymbol{\theta}_t) - \eta \, \mathbf{g}_{\text{val},t}^{\top} \mathbf{H}_{\text{val},t}^{\dagger} \mathbf{g}_{S,t} \; + \; \mathcal{O}(\eta^2) \; + \; o(\eta^2), \tag{34}$$

where the $\mathcal{O}(\eta^2)$ term collects the quadratic Taylor term and the remainder.

Since $\mathcal{L}_{\text{val}}(\boldsymbol{\theta}_t)$ is independent of $S$, minimizing $\mathcal{L}_{\text{val}}(\widetilde{\boldsymbol{\theta}}_S)$ over $S$ is, up to an $S$-independent constant and higher-order terms in $\eta$, equivalent to maximizing the first-order predicted decrease term $\mathbf{g}_{\text{val},t}^{\top} \mathbf{H}_{\text{val},t}^{\dagger} \mathbf{g}_{S,t}$. Formally, ignoring $\mathcal{O}(\eta^2)$ and $o(\eta^2)$ terms in (34), we obtain the surrogate outer problem

$$\max_{S \subseteq \mathcal{D}, \, |S|=k} \mathbf{g}_{\text{val},t}^{\top} \mathbf{H}_{\text{val},t}^{\dagger} \mathbf{g}_{S,t}, \tag{35}$$

which is exactly (7) after substituting the definitions of $\mathbf{g}_{\text{val},t}$ and $\mathbf{g}_{S,t}$.

Finally, because $\widetilde{\boldsymbol{\theta}}_S$ is a first-order local surrogate of the inner solution $\boldsymbol{\theta}_S^*$ around $\boldsymbol{\theta}_t$ (Eq. (30)), this yields the claimed single-level approximation to Problem 1, up to $S$-independent constants and higher-order terms. $\square$

### E.2. Proof of Theorem 3.2

**Theorem 3.2** (LoRA induces cross-block curvature). *Let $\mathcal{L}(W)$ be twice differentiable and consider the LoRA parameterization $W = W_0 + BA$, where $B \in \mathbb{R}^{m \times r}$ and $A \in \mathbb{R}^{r \times n}$. Let $\mathbf{G}_W = \nabla_W \mathcal{L}(W)$ and let $\mathbf{H}_W$ denote the Hessian with respect to $\text{vec}(W)$. For any $i \in [m]$, $j \in [n]$, and $k, k' \in [r]$,*

$$\frac{\partial^2 \mathcal{L}}{\partial B_{ik'} \partial A_{kj}} = \left\langle \mathbf{H}_W[B_{:k} e_j^{\top}], \, e_i A_{k':} \right\rangle_F + \delta_{kk'} (\mathbf{G}_W)_{ij}.$$

*Thus, the LoRA-parameter Hessian contains explicit cross-block terms between $A$ and $B$. In particular, when $k = k'$, the mixed derivative between $B_{ik}$ and $A_{kj}$ contains the additional term $(\mathbf{G}_W)_{ij}$, which arises directly from the bilinear parameterization $BA$. Whenever the right-hand side of Eq. (9) is nonzero for some $(i, j, k, k')$, the LoRA-parameter Hessian has a nonzero cross-block entry that cannot be represented by a diagonal preconditioner.*

*Proof.* Let

$$f(A, B) \triangleq \mathcal{L}(W_0 + BA). \tag{36}$$

We compute the mixed second derivative between $B_{ik'}$ and $A_{kj}$.

First, since

$$\frac{\partial W}{\partial A_{kj}} = B_{:k} e_j^\top, \tag{37}$$

the first derivative with respect to $A_{kj}$ is

$$\frac{\partial f}{\partial A_{kj}} = \left\langle \mathbf{G}_W, B_{:k} e_j^\top \right\rangle_F, \tag{38}$$

where $\mathbf{G}_W = \nabla_W \mathcal{L}(W)$ is evaluated at $W = W_0 + BA$.

Now differentiate this expression with respect to $B_{ik'}$. There are two contributions. The first comes from the dependence of $\mathbf{G}_W$ on $W$, and the second comes from the explicit dependence of $B_{:k} e_j^\top$ on $B$:

$$\frac{\partial^2 f}{\partial B_{ik'} \partial A_{kj}} = \left\langle \frac{\partial \mathbf{G}_W}{\partial B_{ik'}}, B_{:k} e_j^\top \right\rangle_F + \left\langle \mathbf{G}_W, \frac{\partial (B_{:k} e_j^\top)}{\partial B_{ik'}} \right\rangle_F. \tag{39}$$

Since

$$\frac{\partial W}{\partial B_{ik'}} = e_i A_{k':}, \tag{40}$$

the first term can be written as

$$\left\langle \frac{\partial \mathbf{G}_W}{\partial B_{ik'}}, B_{:k} e_j^\top \right\rangle_F = \left\langle \mathbf{H}_W[e_i A_{k':}], B_{:k} e_j^\top \right\rangle_F = \left\langle \mathbf{H}_W[B_{:k} e_j^\top], e_i A_{k':} \right\rangle_F, \tag{41}$$

where the last equality follows from the symmetry of the Hessian.

For the second term, we have

$$\frac{\partial (B_{:k} e_j^\top)}{\partial B_{ik'}} = \delta_{kk'} e_i e_j^\top. \tag{42}$$

Therefore,

$$\left\langle \mathbf{G}_W, \frac{\partial (B_{:k} e_j^\top)}{\partial B_{ik'}} \right\rangle_F = \delta_{kk'} \left\langle \mathbf{G}_W, e_i e_j^\top \right\rangle_F = \delta_{kk'} (\mathbf{G}_W)_{ij}. \tag{43}$$

Combining the two terms gives

$$\frac{\partial^2 \mathcal{L}}{\partial B_{ik'} \partial A_{kj}} = \left\langle \mathbf{H}_W[B_{:k} e_j^\top], e_i A_{k':} \right\rangle_F + \delta_{kk'} (\mathbf{G}_W)_{ij}, \tag{44}$$

which proves Eq. (9).

Combining the two terms gives Eq. (9). The term $\delta_{kk'} (\mathbf{G}_W)_{ij}$ arises from the second derivative of the bilinear map $BA$ with respect to $B_{ik'}$ and $A_{kj}$. Therefore, whenever the right-hand side is nonzero for some $(i, j, k, k')$, the LoRA-parameter Hessian contains a nonzero cross-block entry between $A$ and $B$. $\square$

## E.3. Proof of Theorem 3.3

**Notation and definitions.** A symmetric matrix $\mathbf{M} \in \mathbb{R}^{d \times d}$ is *positive semidefinite (PSD)* if $\mathbf{x}^\top \mathbf{M} \mathbf{x} \geq 0$ for all $\mathbf{x} \in \mathbb{R}^d$. For two $r$-dimensional subspaces $\mathcal{U}, \mathcal{V} \subset \mathbb{R}^d$ with orthonormal bases $\mathbf{U}, \mathbf{V} \in \mathbb{R}^{d \times r}$, the principal angles $\Theta(\mathcal{U}, \mathcal{V}) = \mathrm{diag}(\theta_1, \ldots, \theta_r)$ are defined by $\cos \theta_i = \sigma_i(\mathbf{U}^\top \mathbf{V})$, where $\sigma_i(\cdot)$ denotes singular values. We use $\|\sin \Theta(\mathcal{U}, \mathcal{V})\|_2 = \sin \theta_{\max}$ as the subspace distance.

**Setup.** Let the validation objective be the average negative log-likelihood (NLL)

$$\mathcal{L}_{\mathrm{val}}(\boldsymbol{\theta}) = \frac{1}{n_{\mathrm{val}}} \sum_{i=1}^{n_{\mathrm{val}}} \ell_i(\boldsymbol{\theta}), \qquad \ell_i(\boldsymbol{\theta}) = -\log p_{\boldsymbol{\theta}}(y_i \mid x_i). \tag{45}$$

Denote the validation Hessian $\mathbf{H}_{\mathrm{val},t} = \nabla^2_{\boldsymbol{\theta}} \mathcal{L}_{\mathrm{val}}(\boldsymbol{\theta}_t)$. Let $\mathbf{g}_i(\boldsymbol{\theta}_t) = \nabla_{\boldsymbol{\theta}} \ell_i(\boldsymbol{\theta}_t)$ and $\mathbf{G}_{\mathrm{val},t} \in \mathbb{R}^{d \times n_{\mathrm{val}}}$ stack $\mathbf{g}_i(\boldsymbol{\theta}_t)$ as columns. We define the empirical Fisher-type proxy

$$\widehat{\mathbf{F}}_{\mathrm{val},t} \triangleq \frac{1}{n_{\mathrm{val}}} \mathbf{G}_{\mathrm{val},t} \mathbf{G}_{\mathrm{val},t}^\top. \tag{46}$$

Note that scaling by $1/n_{\mathrm{val}}$ does not affect eigenspaces.

**Assumption E.1** (Eigenspace separation). Let $\lambda_1(\widehat{\mathbf{F}}_{\mathrm{val},t}) \geq \cdots \geq \lambda_d(\widehat{\mathbf{F}}_{\mathrm{val},t}) \geq 0$. Assume the top-$r$ eigenspace of $\widehat{\mathbf{F}}_{\mathrm{val},t}$ is separated by an eigengap

$$\gamma_t \triangleq \lambda_r(\widehat{\mathbf{F}}_{\mathrm{val},t}) - \lambda_{r+1}(\widehat{\mathbf{F}}_{\mathrm{val},t}) > 0. \tag{47}$$

**Lemma E.2** (Gauss–Newton/Fisher decomposition for NLL (Papyan, 2020)). *Consider an NLL objective $\ell(\boldsymbol{\theta}) = -\log p_{\boldsymbol{\theta}}(y \mid x)$. Under standard regularity assumptions (twice differentiability and interchange of derivatives), the Hessian admits the decomposition*

$$\nabla^2_{\boldsymbol{\theta}} \ell(\boldsymbol{\theta}) = \mathbf{F}(\boldsymbol{\theta}) + \mathbf{R}(\boldsymbol{\theta}), \tag{48}$$

*where $\mathbf{F}(\boldsymbol{\theta})$ is a Fisher/Gauss–Newton-type PSD curvature term, and $\mathbf{R}(\boldsymbol{\theta})$ collects residual curvature terms involving second derivatives of the model map (e.g., the "non-Gauss–Newton" component). Consequently, for the validation objective,*

$$\mathbf{H}_{\mathrm{val},t} = \mathbf{F}_{\mathrm{val},t} + \mathbf{R}_t. \tag{49}$$

**Lemma E.3** (Davis–Kahan Theorem (Davis & Kahan, 1970)). *Let $\mathbf{A}, \mathbf{B} \in \mathbb{R}^{d \times d}$ be symmetric, and let $\mathcal{S}_r(\mathbf{A})$ and $\mathcal{S}_r(\mathbf{B})$ denote their top-$r$ eigenspaces. Assume $\mathbf{B}$ has an eigengap $\gamma = \lambda_r(\mathbf{B}) - \lambda_{r+1}(\mathbf{B}) > 0$. Then*

$$\left\| \sin \Theta\big(\mathcal{S}_r(\mathbf{A}), \mathcal{S}_r(\mathbf{B})\big) \right\|_2 \leq \frac{\|\mathbf{A} - \mathbf{B}\|_2}{\gamma}. \tag{50}$$

**Theorem 3.3** (Eigenspace stability of the proxy). *Let $\mathcal{S}_r(\mathbf{M})$ denote the top-$r$ eigenspace of a PSD matrix $\mathbf{M}$. Assume Assumption E.1. For NLL objectives, let $\mathbf{H}_{\mathrm{val},t}$ be the validation Hessian and $\widehat{\mathbf{F}}_{\mathrm{val},t} = \frac{1}{n_{\mathrm{val}}} \mathbf{G}_{\mathrm{val},t} \mathbf{G}_{\mathrm{val},t}^\top$. Let $\mathbf{F}_{\mathrm{val},t}$ be a Fisher/Gauss–Newton-type curvature term from Lemma E.2, so that $\mathbf{H}_{\mathrm{val},t} = \mathbf{F}_{\mathrm{val},t} + \mathbf{R}_t$. Define*

$$\varepsilon_t \triangleq \|\mathbf{R}_t\|_2 + \|\mathbf{F}_{\mathrm{val},t} - \widehat{\mathbf{F}}_{\mathrm{val},t}\|_2. \tag{51}$$

*Then the dominant subspaces are close:*

$$\left\| \sin \Theta\big(\mathcal{S}_r(\mathbf{H}_{\mathrm{val},t}), \ \mathcal{S}_r(\widehat{\mathbf{F}}_{\mathrm{val},t})\big) \right\|_2 \leq \frac{\varepsilon_t}{\gamma_t}. \tag{52}$$

*Proof.* By Lemma E.2, we have the decomposition

$$\mathbf{H}_{\mathrm{val},t} = \mathbf{F}_{\mathrm{val},t} + \mathbf{R}_t. \tag{53}$$

Hence

$$\mathbf{H}_{\mathrm{val},t} - \widehat{\mathbf{F}}_{\mathrm{val},t} = (\mathbf{F}_{\mathrm{val},t} - \widehat{\mathbf{F}}_{\mathrm{val},t}) + \mathbf{R}_t. \tag{54}$$

Taking operator norms and applying the triangle inequality yields

$$\|\mathbf{H}_{\mathrm{val},t} - \widehat{\mathbf{F}}_{\mathrm{val},t}\|_2 \leq \|\mathbf{F}_{\mathrm{val},t} - \widehat{\mathbf{F}}_{\mathrm{val},t}\|_2 + \|\mathbf{R}_t\|_2 = \varepsilon_t. \tag{55}$$

Under Assumption E.1, $\widehat{\mathbf{F}}_{\mathrm{val},t}$ has an eigengap $\gamma_t > 0$ at rank $r$. Applying Lemma E.3 with $\mathbf{A} = \mathbf{H}_{\mathrm{val},t}$ and $\mathbf{B} = \widehat{\mathbf{F}}_{\mathrm{val},t}$ gives

$$\big\| \sin\Theta\big(\mathcal{S}_r(\mathbf{H}_{\mathrm{val},t}),\, \mathcal{S}_r(\widehat{\mathbf{F}}_{\mathrm{val},t})\big)\big\|_2 \leq \frac{\|\mathbf{H}_{\mathrm{val},t} - \widehat{\mathbf{F}}_{\mathrm{val},t}\|_2}{\gamma_t} \leq \frac{\varepsilon_t}{\gamma_t}, \tag{56}$$

which proves the claim. $\qquad\square$

*Remark* E.4 (Theoretical Necessity of Warmup). Theorem 3.3 establishes that the fidelity of our subspace recovery is bounded by the ratio $\varepsilon_t/\gamma_t$. At initialization ($t = 0$), the high negative log-likelihood implies a large non-Gauss-Newton residual term $\|\mathbf{R}_t\|_2$ (inflating $\varepsilon_t$) and a chaotic gradient spectrum with a negligible eigengap $\gamma_t$. Therefore, a lightweight warmup is not merely a heuristic but a theoretical prerequisite. It drives the optimization trajectory into a local basin where the residual curvature decays ($\|\mathbf{R}_t\|_2 \to 0$, validating the Gauss-Newton approximation $\mathbf{H} \approx \mathbf{F}$) and the intrinsic task-specific structure emerges (maximizing $\gamma_t$). This ensures GIST operates in a regime where the empirical proxy is mathematically guaranteed to align with the true Hessian geometry.

# F. Baseline Details

In this section, we provide detailed implementation specifications for all baseline methods compared in our experiments. We categorize these methods into four groups: Random Selection, Hardness-aware Heuristics, Similarity-based Methods, and Optimizer-based Methods. For baseline approaches that involve stochasticity, we perform three runs with different random seeds and report the average performance and standard deviation.

## F.1. Random Selection

We employ **Random Selection** as a standard baseline, which uniformly samples data points from the candidate pool $\mathcal{D}$ until the target budget $k$ is met. Despite its simplicity, random selection often serves as a strong baseline in instruction tuning.

**Random.** We uniformly sample data points from the entire candidate pool $\mathcal{D}$ until the budget is met.

## F.2. Hard Example Mining

These methods select data based on intrinsic properties of the examples (e.g., difficulty or length), independent of the specific target task. In our geometric view (Section 3.1), these methods typically prioritize gradient magnitude over direction.

**Length.** We sort examples by length (in tokens) and take the longest samples. This has been shown to be a strong baseline by Zhao et al. (2024a), and is computationally cheap, only requiring computing the length of each sample in our data pool.

**Perplexity (PPL).** We compute the loss (perplexity) of each sample $d \in \mathcal{D}$ using the pre-trained base model, following prior work (Yin & Rush, 2025; Antonello et al., 2021; Marion et al., 2023; Ankner et al., 2025). Consistent with Yin & Rush (2025), we select samples with the highest loss values (hardest samples).

## F.3. Similarity-based Methods

These methods select training samples $d \in \mathcal{D}$ that are semantically close to the target validation examples $v \in \mathcal{D}_{\mathrm{val}}$.

**Embedding.** Following standard practices in Ivison et al. (2025), we employ the **GTR-Base** model (Ni et al., 2022) as the external encoder. For every candidate sample $d$ and validation sample $v$, we compute their embeddings using GTR-Base and calculate the cosine similarity score. The final score for a candidate $d$ is its maximum similarity to any example in the validation set.

**RDS+.** We adopt the **RDS+** method proposed by Ivison et al. (2025). Unlike standard embedding approaches, RDS+ utilizes the base model's own internal representations to measure similarity. Following Ivison et al. (2025), we extract the hidden states from the last layer and apply a position-weighted mean pooling to derive the representation for each sequence. This weighted pooling strategy is designed to better capture the instruction-following semantics compared to standard mean

pooling. We use the official codebase provided by the authors[2] for implementation.

### F.4. Optimizer-based Methods

These methods estimate the gradient-based influence of training samples derived from optimizer statistics (e.g., Adam's second-moment estimates on the validation loss.

**LESS.** We follow the procedure outlined in Xia et al. (2024). We first train an auxiliary LoRA model on a random subset of the data. Then, we compute the influence score for each pair $(v, d)$ using gradient projections. Crucially, LESS utilizes the optimizer states (Adam's moments) to re-scale gradients. We use the official codebase provided by the authors[3] for implementation. We follow the paper's default setup, using 4 epochs and setting the random projection dimension to 8192. Also we adopt the reported performance of LESS on Llama2-7B from the original paper.

## G. Training

### G.1. Training Datasets

We use the same four preprocessed training datasets as in Wang et al. (2023). All are human-written or human-annotated; details are provided in Table 6. FLAN V2 and COT are derived from existing NLP benchmarks, whereas DOLLY and OPEN ASSISTANT 1 contain open-ended generation examples with human-written responses. These datasets differ substantially in format, length, and task type, highlighting the diversity of instruction-tuning data. Following Wang et al. (2023), we standardize them using the same "Tulu" format.

> **<|user|>**
> Why can camels survive for long without water?
>
> **<|assistant|>**
> Camels use the fat in their humps to keep them filled with energy and hydration for long periods of time.

*Table 6.* Details of training dataset from Wang et al. (2023). Len. is short for token length.

| Dataset | # Instance | Sourced from | # Rounds | Prompt Len. | Completion Len. |
|---|---|---|---|---|---|
| FLAN V2 | 100,000 | NLP datasets and human-written instructions | 1 | 355.7 | 31.2 |
| COT | 100,000 | NLP datasets and human-written CoTs | 1 | 266 | 53.2 |
| DOLLY | 15,011 | Human-written from scratch | 1 | 118.1 | 91.3 |
| OPEN ASSISTANT 1 | 55,668 | Human-written from scratch | 1.6 | 34.8 | 212.5 |

### G.2. Infrastructure and Implementation.

All experiments were conducted on a system equipped with NVIDIA A100 GPU and AMD EPYC 7473X 24-core processor (48 threads).

### G.3. Training Details

We adopt the training configuration from Xia et al. (2024). We implemented parameter-efficient fine-tuning across all experiments using LoRA (Hu et al., 2022). Training was performed for 4 epochs with a batch size of 128, utilizing a learning rate scheduler that combines linear warm-up with cosine decay, peaking at $2 \times 10^{-5}$. For the LoRA configuration, we targeted all attention matrices with a rank of 128, $\alpha = 512$, and a dropout rate of 0.1. This setup resulted in 135M trainable parameters (1.95%) for Llama2-7B, 73M (2.23%) for Llama3.2-3B, and 34M (2.21%) for Qwen2.5-1.5B. To ensure robustness, each experiment was repeated over three trials using distinct random seeds. For random selection baselines, this involved sampling three unique subsets from the training data. For our proposed method, distinct subsets were selected from separate models that had undergone warmup training on varied data partitions. Optimization seeds remained consistent throughout all experiments.

---

[2]https://github.com/hamishivi/automated-instruction-selection
[3]https://github.com/princeton-nlp/LESS

## H. Evaluation

We follow Wang et al. (2023) to evaluate the performance of the models on the target tasks. For MMLU, we measure the 5-shot accuracy of the test set averaged across 57 subtasks. For TYDIQA, we measure the 1-shot macro-averaged F1 score across all 11 languages. We adopt the gold-passage setup where one passage containing the reference answer is provided to the model. For BBH, we report the average 3-shot exact match score across all tasks. Chain-of-thought reasoning is provided in each in-context learning example to prompt the model to generate chain-of-thought reasoning traces for test examples. We evaluate on the validation set $\mathcal{D}_{\text{val}}$ (the same reference set used for data selection) at the end of each epoch and select the best checkpoint to evaluate on the final test set for each experiment. Note that this procedure might introduce some bias to the final test set, given that the validation set is relatively small (e.g., TYDIQA only has 9 validation examples in total). According to Xia et al. (2024), this bias doesn't affect the comparisons between different methods.

## I. More Experiment Results

### I.1. Gradient Spectrum Analysis

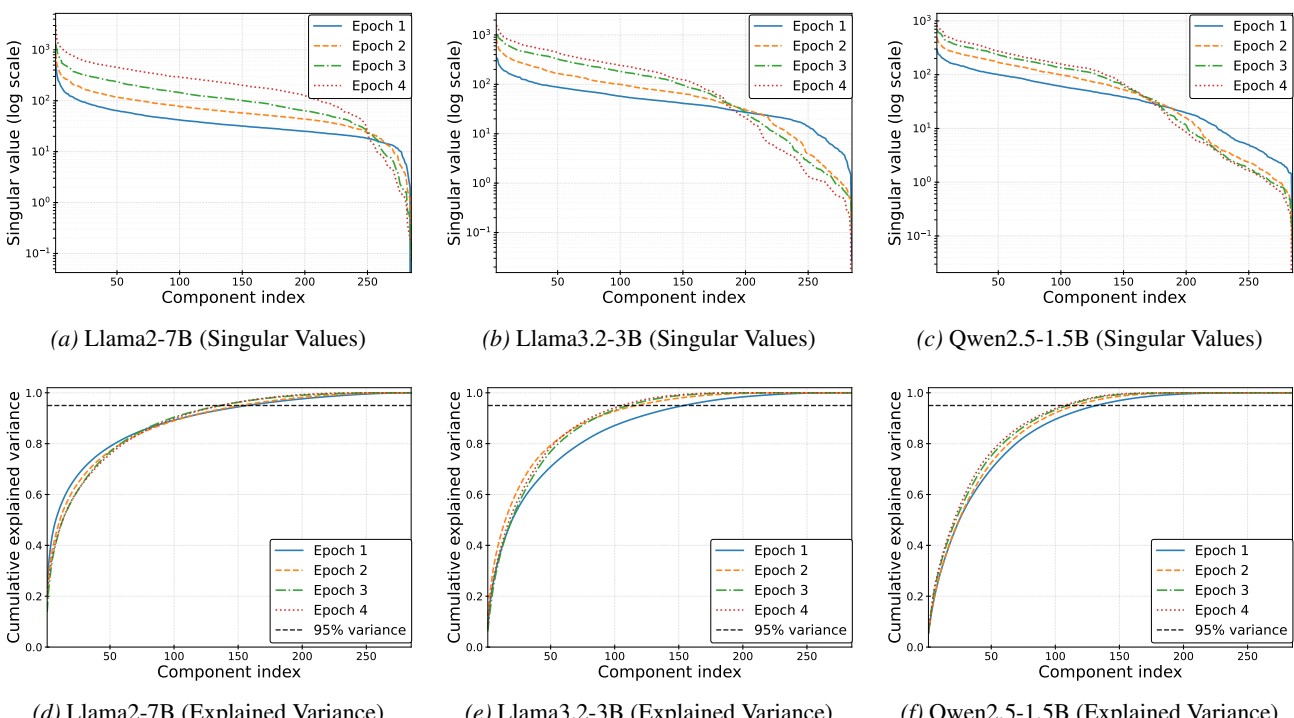

*(a)* Llama2-7B (Singular Values)  *(b)* Llama3.2-3B (Singular Values)  *(c)* Qwen2.5-1.5B (Singular Values)

*(d)* Llama2-7B (Explained Variance)  *(e)* Llama3.2-3B (Explained Variance)  *(f)* Qwen2.5-1.5B (Explained Variance)

*Figure 6.* **Spectral Analysis of Gradient Subspaces across Models and Epochs. Top Row:** Singular value spectra (log scale) of the gradient covariance matrix. Early epochs (blue solid lines) show slower decay, indicating a higher-dimensional optimization landscape. **Bottom Row:** Cumulative explained variance. Later epochs (dotted red lines) reach 95% variance with fewer components, signaling dimensional shrinkage. Notably, larger models (e.g., Llama2-7B) exhibit a slower spectral decay compared to smaller counterparts, reflecting a higher intrinsic dimensionality for task adaptation.

**Early training stages preserve a higher effective dimension, while larger models exhibit richer intrinsic structures.**
Figure 6 visualizes the spectral properties of the target gradient covariance matrices across different training epochs and model architectures. First, we observe that earlier epochs (e.g., Epoch 1) consistently exhibit a "heavier tail" in their singular value distribution compared to later epochs. As training progresses, the spectrum decays more rapidly, and the cumulative variance rises sharper, indicating a collapse of the optimization trajectory into a lower-dimensional subspace. This suggests that the "warm-up" stage contains diverse and exploratory gradient signals crucial for robust data selection, whereas later stages become over-specialized. Regarding model scale, models with larger parameter counts demonstrate a higher intrinsic dimension. By comparing the singular value decay rates, larger models maintain significant singular values across a wider range of components, whereas smaller models show a faster spectral drop-off. This implies that larger models rely on a more complex and high-dimensional feature subspace for task adaptation, necessitating a rank-adaptive selection strategy

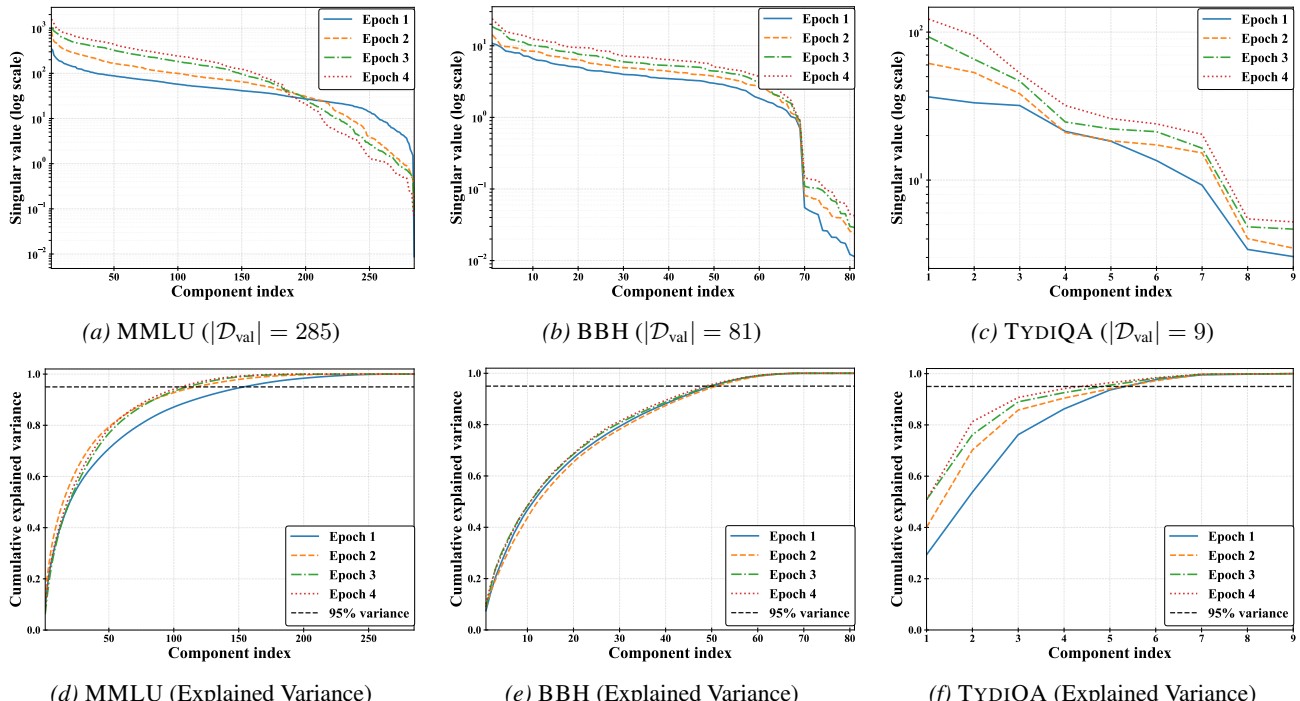

*(a)* MMLU ($|\mathcal{D}_{\text{val}}| = 285$)   *(b)* BBH ($|\mathcal{D}_{\text{val}}| = 81$)   *(c)* TYDIQA ($|\mathcal{D}_{\text{val}}| = 9$)

*(d)* MMLU (Explained Variance)   *(e)* BBH (Explained Variance)   *(f)* TYDIQA (Explained Variance)

*Figure 7.* **Impact of Dataset Scale and Task Type on Gradient Geometry.** We analyze the spectral properties of Llama3.2-3B gradients across three datasets with varying sizes. **Top Row:** Singular value spectra show that intrinsic dimension scales with data size. MMLU exhibits a smooth, heavy-tailed decay, whereas TYDIQA suffers from extreme spectral sparsity due to data scarcity. **Bottom Row:** Cumulative explained variance. Note that across all scales, **Epoch 1 (blue solid lines) consistently preserves a higher effective dimension** (slower variance saturation) compared to later epochs (red dotted lines). This confirms that early-stage gradients are essential for preventing subspace collapse, especially in low-resource regimes like TYDIQA.

like `GIST`.

**Spectral filtering is essential to counteract subspace underspecification in low-resource regimes.** Figure 7 reveals a critical disparity in gradient geometry across data scales. While the data-rich MMLU exhibits a smooth, heavy-tailed singular value distribution (indicating a well-covered feature space), BBH and TYDIQA display marked spectral sparsity. Specifically, for these smaller datasets, the limited sample size is insufficient to span the effective optimization subspace. Unlike MMLU, the eigenvalues of BBH and TYDIQA decrease precipitously, indicating that the empirical geometry is severely rank-deficient. Consequently, the vast majority of the parameter space collapses into the null space of the empirical estimator. In this underspecified regime, any full-rank approximation (like diagonal inversion) would be ill-posed: the variance of the preconditioner would be erroneously dominated by these null-space directions (where curvature is near-zero and inversion explodes), amplifying noise rather than signal. This observation fundamentally justifies `GIST`'s **spectral filtering**: we must rigorously filter for the dominant principal components to recover the valid subspace, explicitly discarding the noisy null space to prevent subspace collapse.

**Adaptive Rank Selection Strategy.** To determine the optimal subspace rank $r$ across diverse tasks, we adopt a **spectral-thresholding strategy with a few-shot safety constraint**. Generally, we define the rank $r$ as the minimum number of components required to capture 95% of the spectral variance of the gradient covariance matrix (calibrated on Llama2-7B). However, for extremely low-resource tasks, we introduce a full-rank retention rule to prevent information loss due to estimation variance. Specifically:

- **MMLU ($r = 150$):** The validation gradients exhibit a heavy-tailed distribution, requiring $\approx 53\%$ of the components (150 out of 285) to reach the 95% variance threshold, confirming the high intrinsic dimensionality of the task.
- **BBH ($r = 50$):** The spectrum saturates more rapidly. A rank of 50 captures the dominant 95% of signals (out of 81), effectively filtering out the tail components associated with optimization noise.
- **TYDIQA ($r = 9$):** Although the 95% variance threshold is met at $r \approx 5$, the absolute sample size ($|\mathcal{D}_{\text{val}}| = 9$) is critically small. In such **extreme few-shot regimes**, the statistical stability of spectral estimation is limited, and the computational

cost of full-rank retention is negligible. Therefore, we override the threshold and retain the full rank ($r = |\mathcal{D}_{\text{val}}| = 9$) to ensure zero information loss.

This hybrid strategy balances theoretical rigor with practical robustness for data-scarce scenarios.

## I.2. Warmup and Training Dynamics of LoRA Finetuning

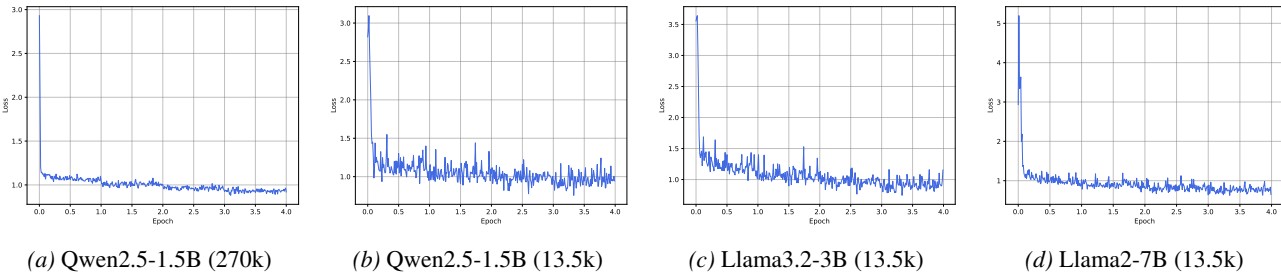

*(a)* Qwen2.5-1.5B (270k)  *(b)* Qwen2.5-1.5B (13.5k)  *(c)* Llama3.2-3B (13.5k)  *(d)* Llama2-7B (13.5k)

*Figure 8.* **Rapid loss convergence in the first epoch creates a stable geometric basin.** We observe consistent training dynamics across varying data scales (13.5k vs. 270k) and model architectures, where the loss drops precipitously within the initial phase ($< 0.5$ epoch) before entering a bounded oscillatory regime.

To verify the assumptions underlying Theorem 3.3, we analyze the training dynamics of multiple instruction-tuned models (Qwen2.5, Llama3.2, Llama2) across different data scales. As illustrated in Figure 8, we consistently observe a distinct two-phase phenomenon: a precipitous drop in loss within the first epoch, followed by a stable oscillatory regime. This behavior is critical for the validity of our spectral approximation. The rapid initial descent effectively minimizes the non-Gauss-Newton residual term $\|\mathbf{R}_t\|_2$ (Eq. (55)), fulfilling the prerequisite for the Fisher information to dominate the Hessian geometry. Furthermore, this transition aligns with the characterization of the Adam optimizer as an "oscillator" in the later stages of training (Cohen et al., 2021; Smith et al., 2018). Once the adaptive second-moment estimates $\hat{\mathbf{v}}_t$ stabilize after the initial shock, the optimization trajectory becomes confined to a low-dimensional basin. In this regime, while the parameters may oscillate locally, the principal subspace of the gradients remains directionally consistent, thereby justifying our strategy of extracting geometric signals from the early-stabilized warmup checkpoints.

## I.3. Cross-Model Transferability

In this section, we ask *whether a subset selected by one model can also benefit other models*. We denote by `GIST-T` the transfer setting where we select data using Qwen2.5-1.5B and then fine-tune Llama2-7B and Llama3.2-3B on the selected subset; results are reported in Table 7. Overall, transfer remains effective and consistently improves over the base model, but it is generally weaker than running `GIST` with the target model. Interestingly, unlike TYDIQA and MMLU, the performance drop on BBH is negligible under transfer.

*Table 7.* Main results with `GIST` and transfer `GIST-T`. `GIST` selects 5% of training data using the target model itself. `GIST-T` selects 5% data *once* using Qwen2.5-1.5B and then reuses the same selected subset to fine-tune other models.

| | **Llama2-7B** | | | | **Llama3.2-3B** | | |
|---|---|---|---|---|---|---|---|
| | **Base** | `GIST` | `GIST-T` | | **Base** | `GIST` | `GIST-T` |
| MMLU | 45.6 | $51.2_{\pm 0.7}$ | $47.0_{\pm 0.4}$ | MMLU | 53.9 | $56.1_{\pm 0.4}$ | $54.5_{\pm 0.1}$ |
| TYDIQA | 46.4 | $55.8_{\pm 0.6}$ | $52.7_{\pm 0.7}$ | TYDIQA | 60.4 | $69.2_{\pm 0.3}$ | $65.0_{\pm 1.4}$ |
| BBH | 38.3 | $41.8_{\pm 0.8}$ | $41.2_{\pm 0.3}$ | BBH | 45.5 | $48.0_{\pm 0.5}$ | $46.8_{\pm 0.6}$ |

## I.4. Ablation of Warmup Stage

Because our analysis in Theorem 3.3 relies on a Newton–Gauss decomposition of the Hessian, we clarify that a brief warmup phase is necessary to drive the loss to a sufficiently small regime. In this section, we empirically validate the importance of warmup. As shown in Table 8, `GIST` with a one-epoch warmup consistently outperforms its no-warmup variant across MMLU, TYDIQA, and BBH.

*Table 8.* Warmup ablation on Llama3.2-3B under the same 5% selection budget. Removing warmup consistently degrades performance across tasks.

| Task | Base | GIST w/o warmup | GIST |
|------|------|-----------------|------|
| MMLU | 53.9 | $53.0_{\pm 0.2}$ | $\mathbf{56.1}_{\pm 0.4}$ |
| TYDIQA | 60.4 | $64.5_{\pm 1.7}$ | $\mathbf{69.2}_{\pm 0.3}$ |
| BBH | 45.5 | $46.2_{\pm 0.4}$ | $\mathbf{48.0}_{\pm 0.5}$ |

## I.5. Influence of LoRA Rank

Table 9 reveals that GIST remains exceptionally robust under a tighter PEFT bottleneck (LoRA rank $r=8$ with 4.6M trainable parameters, accounting 0.14% for Llama3.2-3B), while the baseline LESS degrades noticeably. On TYDIQA, reducing the rank causes LESS to drop from 67.1% to 65.0%, whereas GIST effectively retains its performance (67.5%), outperforming LESS by a clear margin. Remarkably, on BBH, GIST at $r=8$ (48.4%) even slightly exceeds its own $r=128$ result (48.0%), confirming that our selected subsets are highly data-efficient.

*Table 9.* LoRA rank ablation on Llama3.2-3B (5% selection budget). We compare the robustness of GIST against LESS under different LoRA rank constraints. Our method (GIST) maintains significant gains even in the low-rank setting ($r=8$). Gray scripts denote standard deviations across 3 runs.

| Dataset | Base | Random | Default Setting (LoRA $r=128$) | | Low-rank Ablation (LoRA $r=8$) | |
|---------|------|--------|------|------|------|------|
| | | | LESS | GIST | LESS | GIST |
| MMLU | *53.9* | $53.2_{\pm 0.6}$ | $\mathbf{56.5}_{\pm 0.6}$ | $56.1_{\pm 0.4}$ | $\mathbf{56.4}_{\pm 0.2}$ | $55.2_{\pm 0.3}$ |
| TYDIQA | *60.4* | $64.1_{\pm 0.4}$ | $67.1_{\pm 0.8}$ | $\mathbf{69.2}_{\pm 0.3}$ | $65.0_{\pm 0.9}$ | $\mathbf{67.5}_{\pm 0.6}$ |
| BBH | *45.5* | $45.1_{\pm 0.2}$ | $46.1_{\pm 0.7}$ | $\mathbf{48.0}_{\pm 0.5}$ | $46.1_{\pm 0.6}$ | $\mathbf{48.4}_{\pm 0.3}$ |
| *Average* | *53.3* | 54.1 | 56.6 | **57.8** | 55.8 | **57.0** |

We hypothesize that **reducing LoRA rank amplifies parameter coupling**: updates are constrained to a narrower bilinear subspace, making the optimization geometry more "rotated" and less axis-aligned. Methods relying on diagonal surrogates (like LESS) struggle to represent this coupled geometry and thus become sensitive to rank reduction. In contrast, GIST scores examples via alignment within a task-specific subspace extracted from validation gradients. This approach inherently preserves cross-parameter structures, allowing GIST to identify high-value training signals that remain effective even when the trainable degrees of freedom are heavily restricted.

## I.6. Influence of Tail Principle Directions

Table 10 compares GIST against a tail-only variant (GIST-tail) under the same 5% selection budget. While, in optimization view, directions associated with the smallest eigenvalues (tail components) can in principle yield the largest instantaneous decrease in the validation loss, they are also the most vulnerable to overfitting the validation signal and to stochastic gradient noise.

*Table 10.* Tail ablation on Llama3.2-3B under the same 5% selection budget. While GIST consistently improves over *Base* and *Random*, selecting only the smallest principal direction (GIST-tail) can be unstable and may hurt task performance.

| Task | Base | Random | GIST | GIST-tail |
|------|------|--------|------|-----------|
| MMLU | 53.9 | $53.2_{\pm 0.6}$ | $56.1_{\pm 0.4}$ | $\mathbf{56.4}_{\pm 0.2}$ |
| TYDIQA | 60.4 | $64.1_{\pm 0.4}$ | $\mathbf{69.2}_{\pm 0.3}$ | $63.1_{\pm 1.5}$ |
| BBH | 45.5 | $45.1_{\pm 0.2}$ | $\mathbf{48.0}_{\pm 0.5}$ | $38.5_{\pm 1.8}$ |

Empirically, GIST is consistently strong across tasks, whereas GIST-tail exhibits highly task-dependent behavior: it is competitive on MMLU (56.4 vs. 56.1), but drops substantially on TYDIQA (63.1 vs. 69.2) and collapses on BBH (38.5 vs. 48.0). These results suggest that relying only on the decrease of validation loss is brittle, and motivate *automatically* selecting (or weighting) principal directions in a task-adaptive manner, which we leave as an important direction for future

work.

## J. Target Set Subspace Analysis

Since the task projector for TYDIQA in our experiments maps original gradients to a low-dimensional subspace (specifically, a 9-dimensional space), we employ TYDIQA on Qwen2.5-1.5B as a tractable case study to analyze the semantic distribution of subsets selected along each principal direction.

Specifically, utilizing the compact SVD from Eq. (13), we decompose the Gram matrix as $\mathbf{G}_{\text{val},t}^\top \mathbf{G}_{\text{val},t} = V_r \Sigma_r^2 V_r^\top$. As established in Section 3.3, we adopt this matrix as a first-order surrogate for the Hessian, implying $H_{\text{val},t} \approx V_r \Lambda V_r^\top$ (where $\Lambda \approx \Sigma_r^2$). Consequently, the pseudoinverse $H_{\text{val},t}^\dagger$ within the influence function (Eq. (7)) is approximated by $V_r \Lambda^{-1} V_r^\top$.

From an optimization perspective, this spectral analysis highlights a **critical trade-off** between theoretical acceleration and empirical robustness. The leading principal directions (associated with larger singular values of $\mathbf{G}_{\text{val},t}$) correspond to **dominant consensus patterns** with a high signal-to-noise ratio. While these directions ensure reliable descent, they typically contribute to smaller step updates due to high curvature constraints. Conversely, the trailing directions theoretically offer the potential for larger loss reductions (attributable to the inverse scaling effect of $\Lambda^{-1}$ in the influence approximation). However, in practice, these directions are often dominated by **stochastic gradient noise** and estimation errors, rendering them unreliable for generalization despite their theoretical efficiency.

Table 11 illustrates the most influential training examples retrieved along distinct principal directions for a sample Telugu QA instance from the validation set. The English translation of the Telugu query is provided below:

---

<|**user**|>
Context: Pātapāḍēru is a village in Paderu Mandal, Visakhapatnam district. It is 0 km from the mandal headquarters Paderu and 76 km from the nearest town Anakapalli. According to the 2011 Indian Census, the village has 830 households, a population of 3,687, and covers 268 hectares. The number of males is 1,398 and the number of females is 2,289. The Scheduled Castes population is 26, and the Scheduled Tribes population is 2,944. The census location code is 584655. PIN code: 531077.
Question: How many females were there in Pātapāḍēru village in 2011?

<|**assistant**|>
2,289

---

Surprisingly, Table 11 reveals that the samples selected by individual principal directions of $\mathbf{G}_{\text{val},t}$ exhibit a distinct hierarchy of task levels. As previously discussed, leading principal directions represent high-confidence descent directions that, while contributing to smaller loss reductions, align with fundamental and general-purpose language tasks. For instance, PC1 retrieves sentiment analysis reviews, while PC2 selects multi-step multiple-choice questions (MCQs), indicative of general reasoning capabilities.

In contrast, trailing directions target more granular and specific tasks, such as fact-checking (PC8) and non-English sentences (PC9). While incorporating data along these directions yields larger loss reductions on the validation set, these directions are also more prone to falling into the null space. Consequently, such aggressive loss reductions may stem from overfitting, potentially detrimental to generalization. We observe that these semantic patterns across different principal directions remain consistent among the top-scoring samples.

Ultimately, when aggregating all principal directions for selection, the retrieved data semantically converges toward the validation example, heavily favoring Natural Language Inference (NLI) tasks. It also captures specific translation examples; notably, sample (4) `flan_v2_79833` explicitly selects an English-to-Telugu translation pair.

*Table 11.* We use one Telugu TYDIQA validation instance to retrieve the most relevant training examples under GIST. Target Projector (nine principal directions) is precomputed via SVD on the original target set and reused here. For each direction, candidate examples are scored by their projection-based alignment, and the highest-scoring examples are retrieved. We then aggregate the nine directional scores to obtain a single overall score per example, and report the **Top-5** examples with the largest aggregated scores.

---

### A Telugu QA Example in Validation Set

*Multilingual QA in Telugu*

**User:** Context: పాతపాడేరు, విశాఖపట్నం జిల్లా, పాడేరు మండలానికి చెందిన గ్రామము.[1] ఇది మండల కేంద్రమైన పాడేరు నుండి 0 కి. మీ. దూరం లోను, సమీప పట్టణమైన అనకాపల్లి నుండి 76 కి. మీ. దూరంలోనూ ఉంది. 2011 భారత జనగణన గణాంకాల ప్రకారం ఈ గ్రామం 830 ఇళ్లతో, 3687 జనాభాతో 268 హెక్టార్లలో విస్తరించి ఉంది. గ్రామంలో మగవారి సంఖ్య 1398, ఆడవారి సంఖ్య 2289. షెడ్యూల్డ్ కులాల సంఖ్య 26 కాగా షెడ్యూల్డ్ తెగల సంఖ్య 2944. గ్రామం యొక్క జనగణన లొకేషన్ కోడ్ 584655[2].పిన్ కోడ్: 531077.
Question: 2011లో పాతపాడేరు గ్రామంలో ఎంతమంది స్త్రీలు ఉన్నారు?

**Assistant:** 2289

---

| **PC1** | **PC2** | **PC3** |
|---|---|---|
| *PC1: Review Sentiment Classification* | *PC2: Multi-step Reasoning (MCQ)* | *PC3: Common-sense Contradiction* |
| **User:** Title: It's a cute movie!! Review: *It Takes Two* is a cute movie that is loosely based on *The Parent Trap* but also seems loosely based on *The Prince and the Pauper.* Mary-Kate and Ashley are very good and so are Kirstie Alley and Steve Guttenberg. This movie is a lot of fun to watch and worth buying or renting. Question: Is the review positive or negative? | **User:** Given a multiple-choice question, reason step by step and output the selected option. Example: Question: What describes something that requires activation energy to get started? Options: (A) igniting fuel and oxidiser ...(G) chemical equations ...(H) Firecrackers Let's think first: All chemical reactions require activation energy to get started. Chemical equations describe chemical reactions. ... So the answer is [(G)]. ... (additional questions omitted) | **User:** See question below:Choose the sentence that goes against common sense.Options:- Sentence A: "she's asleep, we should keep noisy"- Sentence B: "she's asleep, we should keep quiet" Reason slowly and give your answer. |
| **Assistant:** Positive | | **Assistant:** If you are loud when someone is asleep you will wake them up.The final answer: Sentence A. |

**Assistant:** ... So the answer is **[(H)]**.

---

| **PC4** | **PC5** | **PC6** |
|---|---|---|
| *PC4: Word Game (Scrabble)* | *PC5: Multilingual Mixed Task (Spanish)* | *PC6: Multilingual Reasoning* |
| **User:** I'm playing Scrabble with my family. My seven letters are: A P N W Q V E. What words can I spell with those? | **User:** ¿Que sabes sobre el integrador de Mulesoft? ¿Es mejor plataforma de integración que SAP PI? | **User:** Given the following reasoning and answer, what was the question? Reasoning: Tom served on Monday to $8 \times 10 = 80$ clients. So Tom earned $20/100 \times 80 = 16$ bonus points. Answer: 16 |
| **Assistant:** APE, AWE, ANEW, PAW, PEW, PAN, PEN, PEA, PAVE, PAWN, PANE, PEAN, NAP, NEW, NAVE, NAPE, VAN, VAPE, EW | **Assistant:** MuleSoft is an integration software company that provides the Anypoint platform to connect applications, data, and devices. SAP PI is SAP's tool for centralizing information exchange between SAP and non-SAP systems. They have similar goals but different focuses; the choice depends on the client's needs and what they want to develop. | **Assistant:** In a company, an employee earns bonus points equal to 20% of the number of customers served. Tom served 10 customers per hour for 8 hours on Monday. How many bonus points did Tom earn? |

---

| **PC7** | **PC8** | **PC9** |
|---|---|---|

| PC7 | PC8 | PC9 |
| --- | --- | --- |

*PC7: Translation (Persian → Hebrew)*

**User:** Translate the Persian sentence into Hebrew.
Input (romanized Persian):
Hālā chetor mitavānīm bā dar nazar gereftan-e raftārhā-ye salāmati be pas-andāz beresīm?

**Assistant:** כעת כיצד :נו יכולים לחשוב על ההתנהגויות ברי:ות כדי לחשוב על לחסכון?

*PC8: Verifiable Fact*

**User:** Decide whether each claim is a factual statement.
Example: "In the film *Who Framed Roger Rabbit* real-life characters interacted with cartoon characters."
Answer: Yes.
Claim: "The space opera *Babylon 5* was produced by Warner Bros."
Answer: Yes.

**Assistant:** Warner Bros ran *Babylon 5* from 1993 to 1998.

*PC9: Non-English Sentence (French)*

**User:** Write a sentence not in English.

**Assistant:** Code canadien du travail – et l'Union des employés du transport local et industries diverses, section locale 931, syndicat requérant, – et Camionnage Millman Inc., Saint-Laurent (Québec), employeur.

---

**Overall (Top-5)**

*Overall: Top-5 Highest-Score Prompts*

**(1) flan_v2_93448 — NLI**

**User:** Test for natural language inference. Premise: A group of young girls is running in a field of yellow flowers. Hypothesis: Young girls are in a field of flowers. Options: yes / no / it is not possible to tell.

**Assistant:** Girls who are running in a field of flowers by implication would have to be in a field of flowers. Therefore, the final answer is yes.

**(2) cot_28147 — NLI question reconstruction**
**User:** User: The man playing music is not imply to being a slow jam. Is the reasoning for what question and answer pair?

**Assistant:** Q: Given the sentence "A man playing his music for a crowd." can we conclude "A crowd gathers while a man plays a slow jam."? Options: yes / it is not possible to tell / no. A: it is not possible to tell.

**(3) flan_v2_78693 — False-belief story QA**
**User:** You will be given a definition of a task first, then some input of the task.Given a story, answer the question about the story. The question is the last sentence in the input. These stories can be difficult due to their length and how each story has at least one of the three following scenarios: the first is when the individual's belief matches reality, the second is when the individual's belief does not match reality, and the third is when an individual has a false belief about another individual's beliefs. The question will ask about the location of an object in the story with respect to either none or one of the three scenarios. Note that there are distractor sentences in each story that are unrelated to the question and are designed to confuse the reader. Logan entered the living_room. Ethan entered the living_room. The banana is in the red_crate. Logan moved the banana to the red_cupboard. Logan entered the cellar. Avery entered the cellar. The pumpkin is in the red_envelope. Logan moved the pumpkin to the blue_suitcase. Phone rang. Abigail entered the kitchen. Ethan entered the kitchen. The cabbage is in the green_cupboard. Abigail moved the cabbage to the red_drawer. Logan is in the cellar. Abigail entered the cellar. Phone rang. The pumpkin is in the blue_suitcase. Logan moved the pumpkin to the red_envelope. Where does Logan think that Abigail searches for the pumpkin? Output:

**Assistant:** red_envelope

**(4) flan_v2_79833 — English → Telugu**

**User:** Teacher: A text is given in English. Translate it from the English language to the Telugu language. The translation must not omit or add information to the original sentence. Teacher: Now, understand the problem? Solve this instance: 2017. There are 37,919 Establishments in the the Telangana one. The number of Subscribers stood at 84. Student:

**Assistant:** తెలంగాణ జోన్ పరిధిలో 37,919 సంస్థలు ఉన్నాయని, పిఎఫ్ చందాదారుల సంఖ్య 84.

**(5) flan_v2_7393 — Persian → Hebrew**
**User:** Translate the Persian sentence into Hebrew.
Input (romanized Persian):
Hālā chetor mitavānīm bā dar nazar gereftan-e raftārhā-ye salāmati be pas-andāz beresīm?

**Assistant:** כעת כיצד :נו יכולים לחשוב על ההתנהגויות ברי:ות כדי לחשוב על לחסכון?

