# OpenReview forum: "GIST: Targeted Data Selection for Instruction Tuning via Coupled Optimization Geometry"
_ICML.cc/2026/Conference — ICML 2026 regular_

### Official Review · Reviewer_eT71 · 2026-02-18

**Soundness:** 1
**Presentation:** 3
**Significance:** 1
**Originality:** 2
**Overall Recommendation:** 3
**Confidence:** 5

**Summary:**

The proposed GIST recovers a task-specific subspace from validation gradients via SVD. The GIST projects training gradients into this coupled subspace and scores examples by their alignment with target directions.

**Compliance With Llm Reviewing Policy:**

Affirmed.

**Final Justification:**

As I have replied in rebuttal ack, the authors failed to resolve my concerns regarding several core issues, e.g., novelty and empirical evaluations. Thus, I am unable to support acceptance at this stage and maintain a reject recommendation.

**Key Questions For Authors:**

See weakness.

**Limitations:**

yes

**Strengths And Weaknesses:**

- Strength:
1. The motivation is clear and well-established.
2. The authors provided theoretical analyses for a better understanding.

- Weakness:
1. 1. The idea of using projected gradient alignment is proposed in prior works (TDDS [a]); thus, the novelty is somewhat limited.
2. The abbreviation is misleading, e.g., SVD in Abstract.
3. It would be better to provide an algorithm box in the paper.
4. What is the matrix size of G_val in Section 4.1? If the matrix is large, the efficiency of SVD in step 2 degrades significantly. More details are needed to clarify this.
5. Since the proposed method calculates the alignment score based on the validation datasets, one concern remains whether the proposed method is sensitive to the choice of the validation set, especially on datasets with a biased distribution. However, no ablation or stability is provided to support this.
6. The evaluation is not solid:
      - The compared baselines are weak. It would be more convincing if the paper compared more recent and stronger data selection methods, such as [b,c].
      -  The performance gains are marginal. As shown in Table 2, on MMLU and TYDIQA, the proposed method achieved lower accuracy than the others. More analyses behind this should be elaborated.
      - There are no ablation studies, making the results not convincing.
More experiments should be incorporated. The authors are suggested to move some theoretical content into the appendix, as well as dataset details (e.g., Table 1).
7. The paper fails to do a comprehensive survey; some recent data selection should be discussed to provide a more comprehensive literature review for readers.

Reference:

- [a] Zhang, Xin, et al. "Spanning training progress: Temporal dual-depth scoring (tdds) for enhanced dataset pruning."
- [b] Xie, Sang Michael, et al. "Data selection for language models via importance resampling."
- [c] Zifan Liu, et al. "TSDS: data selection for task-specific model finetuning."

---

> ### Author Rebuttal · Authors · 2026-03-29
>
> We sincerely appreciate your efforts to review our paper and provide valuable suggestions. Below we address each concern in detail.
>
> ---
> - w1: limited novelty related to TDDS.
> - **R w1: We respectfully point out that TDDS and GIST solve fundamentally different problems (coreset selection vs. targeted transfer) with entirely distinct projections**. TDDS is task-agnostic, projecting training gradients onto the accumulated training gradient for temporal stability. GIST operates within the targeted influence framework: our Extracting step computes SVD of validation gradients to capture cross-parameter coupling geometry, directly solving the diagonal surrogate mismatch in PEFT. The two methods share no common framework in problem formulation, projection target, or scoring objective. The other three reviewers all recognize this originality, with Reviewer nSCP rating it "excellent" (4/4).
>
> ---
> - w2: misleading abbreviation (e.g., SVD in the Abstract).
> - **R w2:  SVD refers to Singular Value Decomposition**; we will spell it out at first mention in the revised abstract.
>
> ---
> - w3: better to provide an algorithm box.
> - **R w3**:  **We will add a formal Algorithm block in the revised version to complement our existing pipeline visualization (Figure 2)**. Reviewer nSCP noted Figure 2 is "easy to understand at a high level"; a pseudo-code block will further enhance reproducibility.
>
> ---
> - w4: Matrix size of G_val in Section 4.1 and the efficiency of SVD.
> - **R w4: The exact size of G_val is |D_val|×d (Table 5), but our Extracting step avoids full-space SVD by operating on a compact |D_val|×|D_val| Gram matrix, making decomposition practically instantaneous (Section 5.5).** While G_val is large (e.g., 285×135M for MMLU on Llama2-7B), we leverage spectral symmetry (Section 5.5) to reduce complexity to O(|D_val|^2×d) (Table 4). For MMLU (|D_val|=285), SVD takes 61.06 seconds. We will add a forward reference in Section 4.1.
>
> ---
> - **w5: Sensitivity to the construction and noise of the small target validation set.**
> - **R w5: GIST's spectral filtering inherently mitigates validation set noise, and our experiments across three benchmarks with validation sizes ranging from 9 to 285 already demonstrate robustness at varying scales.** First, our target-set construction strictly follows the protocol of LESS [1], whose authors verified that small validation sets do not affect relative method comparisons (Appendix B of the original paper). Second, to further stress-test robustness under extreme bias, we conducted a new ablation on TyDiQA (Llama3.2-3B), where the default validation set consists of only 9 multilingual samples. We translated all 9 samples into English, completely eliminating language diversity, yet GIST must still select training data for a multilingual test set:
>
> | Setting | LESS F1 | GIST F1 |
> | :--- | :---: | :---: |
> | Default (multilingual) | 67.1±0.8 | **69.2±0.3** |
> | All-English translation | 65.6±1.7 | **66.9±0.9** |
>
> GIST maintains stable performance and consistently outperforms LESS even when the validation distribution is entirely monolingual, confirming that our subspace captures task-level structure rather than surface-level features.
>
> ---
> - w6: Weak baselines, marginal performance gains, lack of ablation studies, and suggestions to move theoretical content/dataset details.
> - **R w6: We respectfully disagree: our baselines align with 2025 top-venue standards [2,3], and GIST delivers competitive-or-superior performance at a fraction of the cost.** DSIR [b] is already outperformed by our baseline LESS. TSDS [c] is similarity-based, a category covered by more recent RDS+ (2025). Also, reviewers JHfn, nSCP, and bW9V all confirm strong performance; Reviewer bW9V highlights "0.29% storage and 25% compute." Regarding ablations, the manuscript includes rank sensitivity (Fig.4), checkpoint analysis (Fig.3), spectral analysis (App.I.1), and efficiency breakdown (Table 4). Reviewer JHfn summarized these as "fairly complete." We will incorporate rebuttal experiments into the revision and move Table 1 to the appendix while retaining theoretical content in the main text.
>
> ---
> - w7: fails to do a comprehensive survey
> - **R w7: Our manuscript already unifies existing methods under a general framework (Section 3.1 & Appendix B) including classic methods.** We will expand the Related Work in the revised version to include the suggested macro-distribution methods (DSIR, TSDS) and recent advances such as IProX [4], with discussion of how they relate to our targeted influence setting.
>
> [1] Xia, Mengzhou, et al. Less: Selecting influential data for targeted instruction tuning. ICML 2024.
>
> [2] Nikdan, Mahdi, et al. Efficient data selection at scale via influence distillation. NeurIPS 2025.
>
> [3] Zhao, Yang, et al. Beyond similarity: A gradient-based graph method for instruction tuning data selection. ACL 2025.
>
> [4] Chen, Sirui, et al. Influence-Preserving Proxies for Gradient-Based Data Selection in LLM Fine-tuning. ICLR 2026.

---

> > ### Author Rebuttal · Reviewer_eT71 · 2026-04-03
> >
> > I appreciate the additional clarifications and the new validation-bias experiment. The response partially addresses concerns regarding implementation details and validation robustness. However, several core issues remain insufficiently resolved.
> >
> > Regarding limited novelty, the proposed method shares conceptual similarity with TDDS in "projected gradient alignment". While the rebuttal clarifies that TDDS and GIST differ in their problem settings and implementations, it does not fully establish why the proposed projected alignment mechanism constitutes substantial novelty rather than an adaptation of prior ideas to a new setting.
> > More importantly, the concerns about evaluation remain largely open: the rebuttal does not directly explain the lower performance on MMLU and TYDIQA. Thus, more analyses are required to comprehensively analyze the effectiveness of the method. In addition, the responses regarding the ablation study (rank sensitivity, checkpoint analysis) do not isolate the contribution of individual components. For instance, as clearly illustrated in Fig. 2, what is the contribution of individual modules to the final performance (e.g., warmup, spectral filtering, geometric scoring)?
> >
> > Therefore, while the rebuttal is helpful in parts, it does not fully resolve my concerns about novelty and empirical solidity.  Consequently, I am unable to support acceptance at this stage.

---

> > > ### Author Response · Authors · 2026-04-03
> > >
> > > We thank the reviewer for the follow-up. We address the remaining points below.
> > >
> > > **Novelty.** We respectfully note that the reviewer's concern has shifted from a specific claim about TDDS to a broader assertion about "conceptual similarity in projected gradient alignment." TDDS does not perform SVD or any form of subspace decomposition — it simply projects onto the accumulated training gradient direction. GIST's core contribution is using SVD to rotate the parameter geometry and decouple cross-parameter interactions induced by LoRA, a fundamentally different geometric operation with distinct theoretical motivation (Theorems 3.2–3.3). Gradient projection as a generic linear algebra operation appears across many subfields; shared use of it does not constitute shared novelty. We will add a detailed comparison with TDDS in the revised Related Work.
> > >
> > > **Performance on MMLU and TyDiQA.** We respectfully clarify that GIST does not consistently underperform on these benchmarks. On Llama3.2-3B (Table 2), GIST achieves the best results on BBH and TyDiQA. On Llama2-7B, GIST is within variance of LESS on BBH/TyDiQA while substantially outperforming on MMLU. GIST achieves matching or superior performance to LESS with only 0.29% storage and 25% compute. As the other three reviewers confirmed, this efficiency-performance tradeoff is a core contribution.
> > >
> > > **Component-wise ablation.** This requirement was not raised in the original review. Nevertheless, we provide a warmup ablation on Llama3.2-3B (5% budget):
> > >
> > > | Task | GIST w/o warmup | GIST |
> > > | :--- | :---: | :---: |
> > > | MMLU | 53.0±0.2 | **56.1±0.4** |
> > > | TyDiQA | 64.5±1.7 | **69.2±0.3** |
> > > | BBH | 46.2±0.4 | **48.0±0.5** |
> > >
> > > Removing warmup degrades performance across all tasks. Combined with the tail ablation (rebuttal **R q1** for reviewer bW9V) isolating spectral truncation and the rank sensitivity analysis (Figure 4) validating the scoring mechanism, each component's contribution is now empirically established. Regarding Figure 2, the coupled geometry it illustrates not only motivates our method but also directly explains the empirical observations in Figure 3.
> > >
> > > We remain open to any further questions and are happy to provide additional clarifications.

---

### Official Review · Reviewer_bW9V · 2026-03-10

**Soundness:** 3
**Presentation:** 3
**Significance:** 3
**Originality:** 3
**Overall Recommendation:** 4
**Confidence:** 4

**Summary:**

This work studies the data selection problem for targeted instruction fine-tuning, which aims to identify the most effective subset of training data samples from the instruction fine-tuning dataset to maximize the target downstream performance.
- First, this work provides a theoretical analysis that interprets the objective of data selection as evaluating how well the training gradients align with the validation gradients under the geometry of the optimization landscape. Further, it explains that existing data selection methods are approximations of the objective. Especially, the prior work LESS approximates the geometry using diagonal scaling from optimizers like Adam.
- Further, this work shows LoRA’s bilinear parameterization leads to off-diagonal curvature. Thus, it is not captured by existing approximation methods. Empirically, this work shows that validation gradients are in a low-dimensional subspace, where the optimization directions are in a small number of principal components.
- Therefore, this work proposes a method to recover the low-dimensional subspace. This method constructs a proxy matrix from validation gradients and performs singular value decomposition to extract the principal components of this gradient matrix. This work shows a theoretical result that the top eigenspace of this gradient covariance approximates the leading eigenspace of the validation loss landscape. Then, this method evaluates how well each example aligns with the directions and selects the top samples with max similarity scores.
- This work conducts experiments to evaluate the proposed method in instruction tuning. It evaluates on a combined set of 270K instruction-tuning examples, and compares with existing approaches, including RDS+, LESS, and metrics based on perplexity losses and embedding similarities.  The experiments fine-tune three base models (Llama2-7B, Llama3.2-3B, and Qwen2.5-1.5B) with LoRA with 5% selected data, and evaluate the test performance on MMLU, TYDIQA, and BBH benchmarks. Across nearly all models and tasks, the proposed method consistently matches or outperforms the baselines. The results show that the method is more efficient, requiring less storage and computation than LESS, as it only needs a single warmup epoch and performs SVD on validation gradients.

**Compliance With Llm Reviewing Policy:**

Affirmed.

**Final Justification:**

Thank the authors for the rebuttal. My main concerns are resolved. I would like to keep my positive score.

**Key Questions For Authors:**

- As the paper discusses the local geometry of the loss landscape, it would be better to validate whether the proposed approach actually recovers the main eigenvector directions of the Hessian matrix. For example, using the Hutchinson estimator or the Stochastic Lanczos Quadrature algorithm.

- There are many previous works on data/task attribution that consider gradient-based estimation for estimating influence functions/fine-tuning performances on a subset of tasks. It would be better to include a discussion of this line of work, for example [1][2].

[1] Park, Sung Min, Kristian Georgiev, Andrew Ilyas, Guillaume Leclerc, and Aleksander Madry. Trak: Attributing model behavior at scale. ICML 2023
[2] Li, Dongyue, Ziniu Zhang, Lu Wang, and Hongyang R. Zhang. Scalable fine-tuning from multiple data sources: A first-order approximation approach. Findings of EMNLP 2024.

**Limitations:**

Yes, this work discussed its limitations.

**Strengths And Weaknesses:**

### Strengths

- This work provides an interpretation of the data subset selection algorithms as an alignment problem between gradients. It identifies the problem in existing works that they approximate the local geometry using a diagonal Hessian matrix.
- This work proposes a data selection method that recovers the off-diagonal information in the Hessian by constructing a proxy matrix from validation gradients and performing singular value decomposition to extract the principal components of this gradient matrix.
- Results show that the proposed method achieves competitive performance, with only 0.29% of the storage and 25% of the computational time.

### Weaknesses

- The assumptions in the theoretical analysis are not clear. This work has used a local second-order approximation to derive the approximation for the data selection objective. However, it would be better to further discuss what specific assumptions are considered in this analysis and when that is true in instruction fine-tuning.
- What is the computational complexity of the proposed method? The method includes computing the gradients and performing SVD on the gradient matrix. It would be better to precisely quantify the computational complexity, such as in Table 4 of the LESS paper.

---

> ### Author Rebuttal · Authors · 2026-03-29
>
> We sincerely appreciate your efforts to review our paper and provide valuable suggestions. Below we address each concern in detail.
>
> ---
> - w1: Assumptions in the theoretical analysis
> - **R w1: Our derivation rests on standard local convexity and small perturbation assumptions, widely adopted in data attribution [1] and particularly well-justified during fine-tuning.** Unlike chaotic pre-training phases, instruction fine-tuning operates in a stable, locally smooth basin [2] where the second-order Taylor expansion is accurate. Our Warming-up step further ensures parameters are near a local optimum where the Hessian reliably captures curvature. We will make these assumptions explicit after Theorem 3.1 in the revision.
>
> ---
> - w2: Computational complexity.
> - **R w2: The full complexity breakdown is provided in Table 4 and Section 5.5.** Specifically: Warming-up costs one epoch of standard LoRA training; Extracting computes per-sample gradients on $D_{val}$ and performs SVD on the $|D_{val}| \times |D_{val}|$ Gram matrix in $O(|D_{val}|^2 \cdot d)$; Scoring computes projected cosine similarities in $O(|D| \cdot r)$. For MMLU on Llama2-7B, the total pipeline takes ~4.5 GPU hours versus ~18 hours for LESS. We will add a forward reference from the method section to Table 4 for clarity.
>
> ---
> - q1: Validating recovery of Hessian eigenvector directions.
> - **R q1: Theorem 3.3 (Eq. 58) directly bounds the subspace angle between the Hessian eigenspace and our eFIM proxy by $\varepsilon_t/\gamma_t$, where $\gamma_t$ is the eigengap at rank r.** Using eFIM as a Hessian surrogate is the same foundational principle behind widely-adopted optimizers: Adam and AdaGrad use the diagonal of the eFIM for per-parameter rescaling [3]. Our approach retains the top-r eigenspace of the full eFIM instead, preserving off-diagonal coupling that diagonal methods discard. Theorem 3.3 shows that alignment quality is governed by the eigengap $\gamma_t$: top-r directions (large $\gamma_t$) enjoy tight Hessian alignment, while tail directions (vanishing $\gamma_t$) cause the bound to blow up. Our truncation explicitly discards these unreliable directions. We provide two empirical confirmations: first, the rank sensitivity analysis on MMLU (Figure 4) shows stable performance across reasonable r values but degradation when r grows large enough to re-admit noisy directions; second, the tail ablation directly isolates this effect:
>
> | Task   | GIST (Top-$r$) | GIST-tail | $\Delta$ |
> | :----- | :------------: | :-----------------------------: | :------: |
> | TyDiQA |  **69.2±0.3**  | 63.1±1.5 |   -6.1   |
> | BBH    |  **48.0±0.5**  | 38.5±1.8 |   -9.5   |
>
> Both confirm that our proxy recovers curvature directions. We will investigate direct Lanczos-based verification in the extended version.
>
> ---
> - q2: Discussion of data/task attribution works.
> - **R q2:** Thanks for your suggestion. **TRAK [4] is already covered in our Appendix B under Scalable Influence Approximations.** We will expand this section to include Li et al. [5] and other influential works along this line, such as DataInf [6], as well as the latest advances InfDist [7] and IProX [8], providing a more comprehensive discussion of how gradient-based attribution methods relate to GIST.
>
> [1] Koh & Liang. Understanding black-box predictions via influence functions. ICML 2017.
>
> [2] Maddox, Wesley J., et al. A simple baseline for Bayesian uncertainty in deep learning. NeurIPS 2019.
>
> [3] Kunstner, Frederik, et al. Limitations of the empirical Fisher approximation. NeurIPS 2019.
>
> [4] Park, Sung Min, et al. TRAK: Attributing model behavior at scale. ICML 2023.
>
> [5] Li, Dongyue, et al. Scalable fine-tuning from multiple data sources. EMNLP Findings 2024.
>
> [6] Kwon, Yongchan, et al. DataInf: Efficiently estimating data influence in LoRA-tuned LLMs and diffusion models. ICLR 2024.
>
> [7] Nikdan, Mahdi, et al. Efficient data selection at scale via influence distillation. NeurIPS 2025.
>
> [8] Chen, Sirui, et al. Influence-Preserving Proxies for Gradient-Based Data Selection in LLM Fine-tuning. ICLR 2026.

---

> > ### Author Rebuttal · Reviewer_bW9V · 2026-03-31
> >
> > Thanks to the authors for their rebuttal. The authors have provided a point-by-point response to my comments and have provided additional evaluations of the Hessian eigenvalue directions. I think my concerns have been addressed. Thus, I would like to remain positive for this paper.

---

> > > ### Author Response · Authors · 2026-04-03
> > >
> > > We are very grateful for your thorough evaluation and constructive suggestions. Your questions on the theoretical assumptions and Hessian eigenspace recovery directly motivated new analyses that strengthened the paper. We will also expand the discussion of data/task attribution works as suggested. We sincerely appreciate your continued support and will incorporate all discussed refinements in the camera-ready version.

---

### Official Review · Reviewer_nSCP · 2026-03-12

**Soundness:** 3
**Presentation:** 3
**Significance:** 3
**Originality:** 4
**Overall Recommendation:** 4
**Confidence:** 3

**Summary:**

This paper studies targeted data selection for instruction tuning and proposes GIST, a method that replaces diagonal optimizer-based geometry surrogates with a low-rank subspace recovered from validation gradients via SVD. The central claim is that, under PEFT methods such as LoRA, optimizer-state-based diagonal preconditioning fails to capture the true coupled update geometry, whereas a projected subspace alignment objective can better identify useful training examples.

**Compliance With Llm Reviewing Policy:**

Affirmed.

**Key Questions For Authors:**

1. Can the authors provide a more operational interpretation of Theorem 3.3? In particular, is there any empirical proxy for $\epsilon_t$ or subspace mismatch that correlates with when GIST helps most relative to LESS?
2. How much of the observed gain is specific to LoRA? Do the authors expect similar benefits for other PEFT parameterizations or for full-model fine-tuning?

**Limitations:**

yes

**Strengths And Weaknesses:**

S1. targeted instruction-tuning data selection under realistic budget constraints. The paper is well motivated, and the criticism of axis-aligned geometry in optimizer-based selection is technically meaningful
S2. proposed method is simple and relatively elegant. The pipeline in Figure 2 is easy to understand at a high level: warmup, SVD on validation gradients, then projected gradient scoring.
S3. The reported results suggest that GIST is competitive or superior to LESS and several simpler baselines across three benchmarks and three backbone models.

W1.  the scope remains somewhat narrow. The method is strongly motivated by LoRA-induced coupling, and all experiments are conducted in LoRA-based PEFT. This is reasonable, but it leaves unanswered whether the gains extend to other PEFT variants, larger-scale models, or even full fine-tuning regimes where the geometry and storage bottlenecks may look different.
W2. Theorem 3.3 ultimately relies on an approximation bound through an error term $\epsilon_t$ that aggregates residual curvature and proxy mismatch, and the usefulness of the result depends on this quantity being small as well as on a favorable eigen gap. However, these conditions are not operationalized in the main paper, and the theory does not appear to provide a direct ranking guarantee for selected examples. In other words, the analysis supports the plausibility of the proposed proxy, but it falls short of establishing when GIST should reliably outperform diagonal surrogates in practice.

---

> ### Author Rebuttal · Authors · 2026-03-29
>
> We sincerely appreciate your efforts to review our paper and provide valuable suggestions. Below we address each concern in detail.
>
> ---
> - w1: Scope limited to LoRA (applicability to other PEFTs and Full Fine-Tuning).
> - **R w1: While Theorem 3.2 targets LoRA's bilinear coupling, GIST's pipeline extends to any coupled PEFT, and its geometric philosophy aligns with broader LLM optimization trends.** Our pipeline (SVD subspace extraction + projected scoring) applies to any PEFT where cross-parameter coupling renders diagonal geometry suboptimal and to full fine-tuning. This rotation-based correction principle mirrors recent advances in LLM optimizers: GaLore ([1], ICML 2024 Oral) and SOAP ([2], ICLR 2025) periodically apply SVD to rotate the loss landscape, and Muon ([3], arxiv 2025) uses Newton-Schulz iteration for layer-wise orthogonal approximation. GIST applies this same philosophy to gradient-based data selection. For full fine-tuning, we acknowledge that the diagonal mismatch is less severe and relative gains would likely diminish. We will add this discussion to the revised manuscript.
>
> ---
> - w2 & q1: Operational interpretation of Theorem 3.3 and empirical proxy for subspace mismatch.
> - **R w2 & q1: The error term is operationalized through spectral truncation: smaller eigenvalues shrink the eigengap denominator in Eq. (58) and amplify alignment error; we validate this by isolating the tail components.** Using eFIM as a Hessian surrogate is a well-established technique in optimizer rescaling (e.g., Adam's second-moment estimation), but Theorem 3.3 highlights that proxy mismatch is dominated by small-eigenvalue directions where the eigengap vanishes. We provide two empirical confirmations. First, our rank sensitivity analysis on MMLU (Figure 4) shows stable performance across reasonable $r$ values but degradation when $r$ grows large enough to re-admit noisy directions. Second, we conducted a tail ablation on Llama3.2-3B (5% budget), selecting using *only* the smallest principal directions:
>
> | Task   | GIST (Top-$r$) | GIST-tail | $\Delta$ |
> | :----- | :------------: | :-----------------------------: | :------: |
> | TyDiQA |  **69.2±0.3**  | 63.1±1.5 |   -6.1   |
> | BBH    |  **48.0±0.5**  | 38.5±1.8 |   -9.5   |
>
> The sharp degradation confirms that small-eigenvalue directions carry disproportionate surrogate error, validating our truncation design. We acknowledge the theory establishes plausibility rather than a ranking guarantee; tighter distribution-dependent bounds are an interesting future direction.
>
> ---
> - q2: Gains beyond LoRA.
> - **R q2:** Addressed in w1 above. In summary, we expect GIST to benefit any PEFT with cross-parameter coupling, with diminishing gains for full fine-tuning where diagonal mismatch is less severe.
>
> [1] Zhao, Jiawei, et al. Galore: Memory-efficient llm training by gradient low-rank projection. ICML 2024.
>
> [2] Vyas, Nikhil, et al. SOAP: Improving and Stabilizing Shampoo using Adam for Language Modeling. ICLR 2025.
>
> [3] Liu, Jingyuan, et al. Muon is scalable for llm training. arxiv 2025.

---

> > ### Author Rebuttal · Reviewer_nSCP · 2026-04-04
> >
> > Thanks for the detailed responses from the authors. All of my concerns have been addressed. I will maintain my positive score.

---

> > > ### Author Response · Authors · 2026-04-05
> > >
> > > We are very grateful for your rigorous and constructive evaluation. Your questions on the operational interpretation of Theorem 3.3 and the scope beyond LoRA directly motivated new analyses and a broader discussion that strengthened the paper. We sincerely appreciate your continued support and will incorporate all discussed refinements in the camera-ready version.

---

### Official Review · Reviewer_JHfn · 2026-03-13

**Soundness:** 3
**Presentation:** 3
**Significance:** 3
**Originality:** 3
**Overall Recommendation:** 4
**Confidence:** 3

**Summary:**

The paper studies targeted instruction tuning: given a large instruction pool and a small target validation set, select a subset of training data that leads to the best downstream performance on the target task after LoRA fine-tuning. The main claim is that prior  selectors rely on a diagonal approximation that is mismatched to the coupled geometry induced by LoRA. To solve this, he paper proposes GIST, which performs a short LoRA warmup, extracts a low-dimensional target subspace from validation gradients via SVD, and scores candidate examples by gradient alignment inside this subspace. The paper provides theoretical motivation for this construction and shows strong empirical results against other baselines.

**Compliance With Llm Reviewing Policy:**

Affirmed.

**Key Questions For Authors:**

* How sensitive is the proxy in Theorem 3.3 to the size and noise level of the target validation set in practice, especially when $D_{\mathrm{val}}$  is very small  in some cases？

* Does the selected subset transfer across models? For example, if a subset is selected using model A, is it also effective for model B, or is the method fundamentally model-specific?

* If the goal is simply to improve target-task performance for a given model, what is the practical advantage of this targeted selection pipeline over directly adapting on the target task itself using few-shot fine-tuning / LoRA / related methods? In other words, what is the intended application scenario in which this form of target-aware subset selection is preferable?

**Limitations:**

yes

**Strengths And Weaknesses:**

## Strengths

* The motivation and theoretical formulation are clear. The paper identifies a concrete limitation of diagonal geometry in LoRA-based data selection and gives a clean alternative.

* The method is simple and easy to implement. The overall pipeline matches the theoretical discussion well.

* The experiments are fairly complete, and the accompanying analyses are useful. In particular, the result includes efficiency comparisons, spectral analysis, sensitivity analysis.

## Weaknesses

* The target validation sets are quite small in some settings, which makes the selection procedure potentially sensitive to the construction and noise of $D_{\mathrm{val}}$. This is especially relevant because the method explicitly depends on recovering a target subspace from validation gradients.

* The “less is more” claim should be interpreted carefully. The method is explicitly target-aware: it uses target-task information to guide selection. Therefore, it is not very surprising that a target-guided 5% subset can outperform full fine-tuning on a large heterogeneous pool that does not use target information in this way.

---

> ### Author Rebuttal · Authors · 2026-03-29
>
> We sincerely appreciate your efforts to review our paper and provide valuable suggestions. Below we address each concern in detail.
>
> ---
> - w1: Sensitivity to the construction and noise of the small target validation set.
> - **R w1: GIST's spectral filtering inherently mitigates validation set noise, and our experiments across three benchmarks with validation sizes ranging from 9 to 285 already demonstrate robustness at varying scales.** First, our target-set construction strictly follows the protocol of LESS [1], whose authors verified that small validation sets do not affect relative method comparisons (Appendix B of the original paper). Second, to further stress-test robustness under extreme bias, we conducted a new ablation on TyDiQA (Llama3.2-3B), where the default validation set consists of only 9 multilingual samples. We translated all 9 samples into English, completely eliminating language diversity, yet GIST must still select training data for a multilingual test set:
>
> | Setting | LESS F1 | GIST F1 |
> | :--- | :---: | :---: |
> | Default (multilingual) | 67.1±0.8 | **69.2±0.3** |
> | All-English translation | 65.6±1.7 | **66.9±0.9** |
>
> GIST maintains stable performance and consistently outperforms LESS even when the validation distribution is entirely monolingual, confirming that our subspace captures task-level structure rather than surface-level features.
>
> ---
> - w2: "less is more" interpretation
> - **R w2: We completely agree with this insightful interpretation and will reframe "less is more" as a demonstration of mitigating negative transfer.** Our intention was not to claim a paradoxical scaling property, but to highlight that large, heterogeneous pools inherently introduce severe negative transfer for specific downstream tasks. The fact that our target-guided 5% subset outperforms full fine-tuning empirically proves that removing detrimental, out-of-domain data is essential. We will revise the manuscript to explicitly reflect your exact interpretation, clarifying that our method succeeds precisely because it mitigates this negative transfer.
>
> ---
> - q1: Sensitivity of Theorem 3.3's proxy to validation set size.
> - **R q1: The low-rank SVD truncation in our Extracting step acts as a spectral filter against small-sample noise; we verify this by directly measuring proxy stability under reduced $|\mathcal{D}_{\text{val}}|$.** We subsampled the MMLU validation set ($|\mathcal{D}_{\text{val}}|=285$) and measured Spearman's rank correlation and Top-5% selection overlap of the resulting cosine similarity scores against the full-set baseline on Llama3.2-3B:
>
> | D_val Size | Spearman's ρ | Top-5% Overlap (%) |
> | :--------- | :----------: | :----------------: |
> | 100        |0.71|83.8|
> | 50         |0.70 | 84.7|
> | 10         |0.67|78.6|
>
> This directly confirms that the proxy from Theorem 3.3 remains stable even under extreme validation constraints, consistent with the theoretical expectation that top-$r$ truncation discards the long-tail noise amplified by small sample sizes.
>
> ---
> - q2: Does the selected subset transfer across models, or is the method fundamentally model-specific?
> - **R q2: The selected subset shows meaningful cross-model transferability, though GIST is designed to be model-specific since gradients inherently depend on each model's parameter geometry.** To test this, we used Qwen2.5-1.5B to select a 5% subset (GIST-T) and evaluated it on Llama2-7B and Llama3.2-3B. As shown below, GIST-T consistently matches or outperforms the Random baseline, confirming cross-model viability. However, computing alignment within the target model's own subspace remains necessary for optimal performance.
>
> | Target Model | Dataset | Random (5%) | GIST-T (Transfer from Qwen) |
> | :--- | :--- | :---: | :---: |
> | **Llama2-7B** | MMLU | 46.5±0.5 | **47.0±0.4** |
> | | TyDiQA | **52.7±0.4** | **52.7±0.7** |
> | | BBH | 38.9±0.5 | **41.2±0.3** |
> | **Llama3.2-3B** | MMLU | 53.2±0.6 | **54.5±0.1** |
> | | TyDiQA | 64.1±0.4 | **65.0±1.4** |
> | | BBH | 45.1±0.2 | **46.8±0.6** |
>
> ---
> - q3: Practical advantage over direct adaptation on the target task.
> - **R q3: Direct fine-tuning on tiny target sets (e.g., 9 examples in TyDiQA) causes severe overfitting; GIST instead uses them as geometric probes to retrieve diverse training data sharing the same underlying capability.** This shifts the paradigm from memorizing specific target instances to generalizing the target skill via a much larger, automatically curated subset. The intended scenario is any setting where a practitioner has a small specification of a desired capability but access to a large heterogeneous instruction pool. For example, specializing models for distinct roles in multi-agent systems, or customizing a general-purpose model for a domain-specific deployment. GIST automates this data curation without manual subset engineering.
>
> [1] Xia, Mengzhou, et al. Less: Selecting influential data for targeted instruction tuning. ICML 2024.

---

> > ### Author Rebuttal · Reviewer_JHfn · 2026-04-04
> >
> > Thank the authors for the rebuttal. My main concerns are resolved. I would like to keep my positive score.

---

> > > ### Author Response · Authors · 2026-04-05
> > >
> > > We sincerely thank you for your thoughtful and detailed review. Your questions on validation set sensitivity, cross-model transferability, and the practical application scenario directly motivated new experiments that strengthened our empirical contributions. We appreciate your continued support and will incorporate all discussed refinements in the camera-ready version.

---

### Decision · Program_Chairs · 2026-04-30

**Decision:**

Accept (regular)

**Comment:**

The paper introduces GIST, a principled and highly efficient method for targeted data selection in instruction fine-tuning. The authors identify a critical technical gap in existing state-of-the-art methods like LESS: the reliance on diagonal approximations (axis-aligned scaling) that fail to capture the complex, cross-parameter coupling inherent in LoRA-based PEFT. By utilizing SVD on validation gradients to recover a task-specific subspace, GIST scores training examples based on their alignment with target update directions. The work is technically sound, well-written, and offers substantial utility to the community by achieving competitive or superior performance using only 0.29% of the storage and 25% of the computational time of previous baselines. Based on the 1) strong theoretical foundation: the paper provides a meaningful critique of diagonal preconditioning in PEFT. The mathematical derivation of the "coupled optimization geometry" and the use of spectral filtering to mitigate noise in small validation sets are well-grounded; and 2) practical efficiency: the ability to match performance while drastically reducing the storage and compute overhead is a significant contribution that makes targeted data selection accessible for large-scale industrial applications; and 3) methodological clarity: reviewers praised the simplicity and elegance of the pipeline, noting that it is easy to understand and implement,  I recommend this paper for Acceptance.